# A digital twin solution for floating offshore wind turbines validated using a full-scale prototype

Emmanuel Branlard[1], Jason Jonkman[1], Cameron Brown[2], and Jiatian Zhang [2]

[1]National Renewable Energy Laboratory, Golden, CO 80401, USA
[2]Stiesdal Offshore A/S, Denmark

**Correspondence:** E. Branlard (emmanuel.branlard@nrel.gov)

**Abstract.** In this work, we implement, verify, and validate a physics-based digital twin solution applied to a floating offshore wind turbine. The digital twin is validated using measurement data from the full-scale TetraSpar prototype. We focus on the estimation of the aerodynamic loads, wind speed, and section loads along the tower, with the aim of estimating the fatigue lifetime of the tower. Our digital twin solution integrates 1) a Kalman filter to estimate the structural states based on a linear model of the structure and measurements from the turbine, 2) an aerodynamic estimator, and 3) a physics-based virtual sensing procedure to obtain the loads along the tower. The digital twin relies on a set of measurements that are expected to be available on any existing wind turbine (power, pitch, rotor speed, and tower acceleration), and motion sensors that are likely to be standard measurements for a floating platform (inclinometers and GPS sensors). We explore two different pathways to obtain physics-based models: a suite of dedicated Python tools implemented as part of this work and the OpenFAST linearization feature. In our final version of the digital twin, we use components from both approaches. We perform different numerical experiments to verify the individual models of the digital twin. In this simulation realm, we obtain estimated damage equivalent loads of the tower fore-aft bending moment with an accuracy of approximately 5% to 10%. When comparing the digital twin estimations with the measurements from the TetraSpar prototype, the errors increased to 10%–15% on average. Overall, the accuracy of the results is promising and demonstrates the possibility of using digital twin solutions to estimate fatigue loads on floating offshore wind turbines. A natural continuation of this work would be to implement the monitoring and diagnostics aspect of the digital twin to inform operation and maintenance decisions. The digital twin solution is provided with examples as part of an open-source repository.

## 1 Introduction

The offshore floating wind turbine market is expected to grow in the coming decades as the technology gains in maturity, with several floating wind turbine prototypes already tested and commissioned, such as the TetraSpar, developed by Stiesdal Offshore (Stiesdal Offshore, 2022). Operation and maintenance (O&M) costs can account for approximately one-third of offshore wind farm life cycle expenditures for a fixed-bottom project and are expected to be higher for remote (floating) projects (Castella, 2020). Reducing the O&M costs is therefore an impactful and effective means to lower the costs of floating offshore projects. Digital twin solutions are increasingly being used to follow products during their life cycle to assess compo-

nent conditions, guide predictive maintenance, and thereby reduce O&M costs. A review of digital twins for power systems is found in Song et al. (2023). Digital twins often include a virtual sensing component that provides information not measured by the physical system, and a structural health monitoring component to assess the condition of the system. Virtual sensing technology is usually achieved using physics-based or data-driven approaches; both approaches relying on measurements from the physical system to infer and extrapolate information about its current state. Physics-based approaches use a numerical model of the system, whereas data-driven approaches use either ad hoc algorithms or machine-learning techniques. Machine-learning approaches can be trained using high-fidelity models or measurements, leading to potentially high accuracies while maintaining low computational time, but their training requirements imply that a technology cannot be readily transferred from one platform to another. Physics-based models often require low-fidelity models to achieve computational times low enough for digital twins to run in real time. They nevertheless offer the advantage that they provide tractable and insightful results, and, they can be applied to a same family of wind turbine concepts because they do not require a training dataset. Currently, there is no definite case as to which approach can lead to the best digital twin implementation, and it is possible that future approaches will combine physics-based with data-driven techniques. This work presents the development, verification, and validation of a physics-based digital twin for floating wind turbines as a proof of concept for future maturation of the technology.

Digital twins for wind turbine applications have recently become a topic of research interest. The authors explored the topic of physics-based digital twins in previous work, in which a method to estimate tower loads on land-based turbines was developed (Branlard et al., 2020a, b). The approach relied on a Kalman filter model (Kalman, 1960; Zarchan and Musoff, 2015) that combines a linear physics-based model of the structure with measurements from the turbine to perform a virtual sensing of the tower section loads and estimate the fatigue of this component. The measurement data were taken from the supervisory control and data acquisition (SCADA) system using sensors readily available on most turbines. The approach used a mix between an augmented Kalman filter approach (Lourens et al., 2012), where the loads are estimated with the states of the system, and a physics-based aerodynamic estimator for aerodynamic thrust. Bilbao et al. (2022) used a Gaussian process latent force model to estimate the forcing of the system and thereby obtain the section loads along the tower. Drivetrains are another component for which a digital twin has been applied, with physics-based approaches presented in Mehlan et al. (2022, 2023), and data-driven models presented in Kamel et al. (2023).

Despite the recent popularity of the term "digital twin," the concept is heavily based on the fields of structural health monitoring and load estimations (or more generally, virtual sensing), which have long been topics of research. For instance, Iliopoulos et al. (2016) used physics-based modal decomposition to estimate the dynamic response on the substructure of a fixed-bottom wind turbine. Neural networks have been used to establish transfer functions or surrogate models based on SCADA data to obtain wind turbine loads with the aim of performing conditional monitoring (see, e.g., Cosack (2010); Schröder et al. (2018)). Kalman filters were introduced in fields other than wind energy to perform load estimation (Auger et al., 2013; Ma and Ho, 2004; Eftekhar Azam et al., 2015; Lourens et al., 2012). Kalman filtering has been extensively used in wind energy to estimate rotor loads and improve wind turbine control (Boukhezzar and Siguerdidjane, 2011; Selvam et al., 2009; Bottasso and Croce, 2009; Bossanyi, 2003). Load estimations were also achieved using hybrid techniques combining physics based on SCADA data by Noppe et al. (2016). Other load estimation techniques may be used, such as lookup tables (Mendez Reyes et al., 2019),

modal expansion (Iliopoulos et al., 2016), machine learning (Evans et al., 2018), neural networks (Schröder et al., 2018), polynomial chaos expansion (Dimitrov et al., 2018), deconvolution (Jacquelin et al., 2003), or load extrapolation (Ziegler et al., 2017).

In this work, we build on our previous work related to fixed-bottom turbines and present a digital twin solution for floating wind turbines that relies on physics-based models and a Kalman filter. We apply the digital twin to the TetraSpar structure and use measurements from the full-scale prototype. Achieving computational efficiency is crucial to be able to run the digital twin online, therefore, a reduced order model with few selected degrees of freedom is used. Developing digital twins for floating wind turbines present a set of challenges compared to our previous work on fixed-bottom foundations. The potentially large motions undergone by the platform may affect the aerodynamics and accelerometer signals. The models developed for fixed-bottom foundations need to be augmented to be able to predict the aerodynamics when the platform experiences large pitching motions. The dynamics of the platform motion needs to be well captured for the tower-top accelerometer to be used and for estimating the loading in the stationkeeping system. In both floating and fixed-bottom wind turbines, hydrodynamic loads need to be estimated to capture member-level loads in the substructure but they can be omitted as a first approximation if only the tower loads are estimated, as in this study.

In section 2, we provide an overview of our digital concept, the vision for future application, and the TetraSpar prototype on which the digital twin is applied. In section 3, we present the individual components of the digital twin, and run some isolated verification studies on them. In section 4, we present results from the digital twin application first using numerical experiments, then using measurements from the TetraSpar prototype before concluding. To avoid lengthening the main text, we provide derivations (some of which are important contributions of this work) and additional results in appendices.

## 2 Overview of the digital twin concept

In this section, we provide an overview of our digital twin concept and how it is applied in this study.

### 2.1 Long-term vision of the digital twin concept

Many definitions and applications of digital twins are possible. The vision for the concept discussed in this work is to follow the life cycle of a wind turbine in real time and ultimately provide tangible signals to inform O&M decisions. Our goal is to achieve this by relying only on measurements expected to be available on most wind turbines, thereby avoiding the extra cost of adding sensors. In this work, we leave open the question as to whether the installation of an additional set of optimally placed and selected sensors can further improve the predictions of the digital twin, further reducing the long-term O&M costs, and thereby warranting the additional costs of adding the sensors.

We illustrate our approach and vision in Figure 1. The digital twin is intended to run in real time on a cloud platform. It combines a set of models (on the left of the figure) with data from the real system (on the right) to perform the estimation of various states and eventually produce diagnostics that can be use to inform the O&M. The data from the real system are taken from high-frequency measurements from the SCADA system (e.g., power, pitch, rotor speed). The states estimated by the

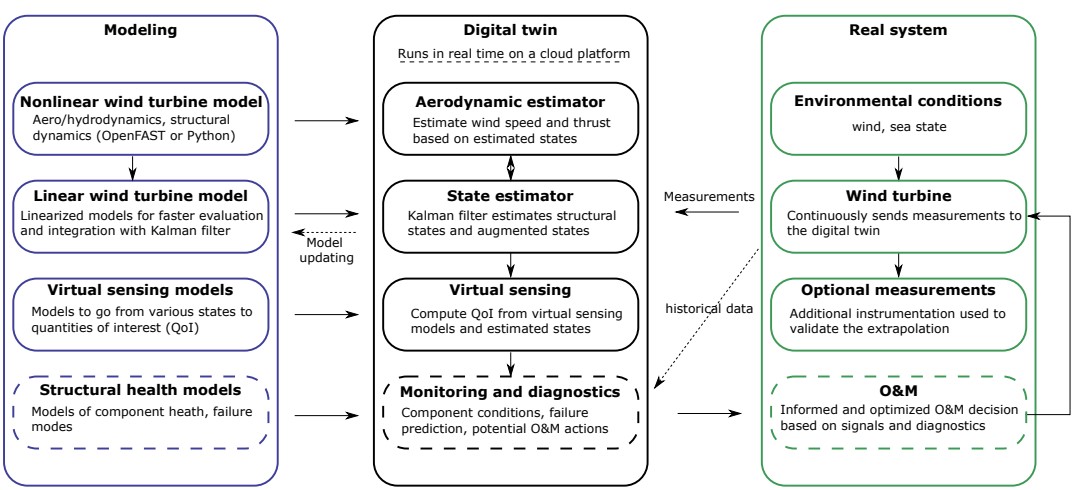

**Figure 1.** Overview of the digital twin concept. Dashed lines indicate features that are outside the current scope.

digital twin include aerodynamic states (wind speed, thrust) and motions of the structure (e.g., surge, pitch, tower deflection).
The core algorithm in the estimation is a Kalman filter that uses a linear wind turbine model. The estimated states are used in
a "virtual sensing" step to produce quantities of interests (QoI), such as the loads at key locations of the structure. The QoI are
then intended to be postprocessed by a monitoring and diagnostic tool to generate the data needed to perform condition-based
O&M.

## 2.2  Narrowed scope

The boxes in Figure 1 with dashed-line borders—structural health modeling, monitoring and diagnostics, and O&M decisions—
are postponed to future work, even though they are essential steps to achieve our final vision. Dashed lines and arrows indicate
options that may be exploited in the future but are also outside of our scope: the use of historical data to assist in the diagnostics,
the use of estimates to perform model updating, and real-time implementation.
This work therefore focuses on the estimation of states and environmental conditions under the assumption that the estimated
quantities can replace costly measurements and eventually be used for O&M decisions. We intend to provide a proof of concept
that paves the way for future commercial applications. A detailed description of each of the boxes surrounded with solid lines
is given in section 3.

## 2.3  System studied

### 2.3.1  The TetraSpar prototype

The system studied for this article is the TetraSpar floating offshore prototype. The system consists of a floating platform
and stationkeeping system developed by Stiesdal Offshore in collaboration with partners Shell, RWE, and TEPCO Renewable
Power, and a 3.6-MW wind turbine with a rotor diameter of 130 m developed by Siemens Gamesa Renewable Energy. A sketch

of the system is provided in Figure 2. The prototype was installed off the coast of Norway and commissioned in November

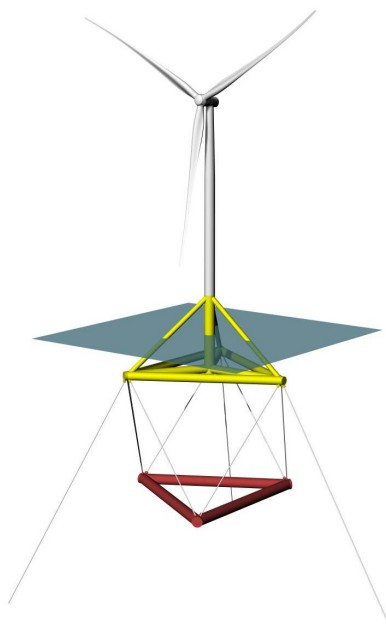

**Figure 2.** Sketch of the TetraSpar prototype


2021. The prototype turbine is equipped with additional sensors (labeled "Optional measurements" in Figure 1), which we use
to validate the estimated QoI.

### 2.3.2 Numerical experiments

Prior to using measurement data, we use simulations (referred to as "numerical experiments") in place of the real system to
feed data to the digital twin. The advantage of this approach is that the QoI are directly accessible and can be compared to the
estimates for verification purposes.
Data for the numerical experiments are obtained using OpenFAST simulations (Jonkman et al., 2023). A model of the
TetraSpar floating platform and the wind turbine was implemented in OpenFAST based on data provided by the manufacturers.
All the members of the substructure are modelled using the strip-theory approach (Morison equation) because the inherent
long-wavelength assumption of the strip-theory has been shown to be sufficiently accurate for this structure with relatively
slender members. The OpenFAST model is complemented with NREL's Reference OpenSource Controller (ROSCO, Abbas
et al. (2021)). The controller parameters are tuned so that OpenFAST simulations match the operating conditions of the turbine
extracted from SCADA data (pitch, rotor speed and power). The nacelle velocity feedback option of ROSCO is used to reduce
the platform pitching motion. Using trial and error, the frequency and damping ratio of the pitch PI-controller are set to
$\omega_p = 0.05$ rad/s and $\zeta_p = 7$ %, and the values for the torque controller are set $\omega_Q = 0.15$ rad/s and $\zeta_Q = 7$ %. The gain-
scheduling of the pitch controller are obtained using the tuning feature of ROSCO. We note that the controller is only needed
to perform verifications of the digital twin with realistic time series of the turbine responses, but the controller itself is not used
for the design of the digital twin. We use the following modules of OpenFAST (Jonkman et al., 2023): MAP (mooring lines),
HydroDyn (hydrodynamics), ElastoDyn (tower and blade elasticity; rigid floater), AeroDyn (aerodynamics), InflowWind (wind
inflow), and ServoDyn (controller interface).
For the numerical experiments, we use synthetic turbulent wind fields generated using TurbSim (Jonkman and Buhl, 2006).
In particular, we often use the same wind field, which we refer to as the "turbulent step," where a deterministic ramp and drop
are added to a turbulent field. The advantage of this 10-min wind field is that it covers all the operating regions of the turbine
in a challenging way because the variations of the wind speed are sudden. The wind speed at hub height for the turbulent step
can be seen in Figure 6.
**2.3.3   Main aspects of the structural model**

We model the structure using a set of 8 degrees of freedom (DOF), as illustrated in Figure 3. The platform is represented as a

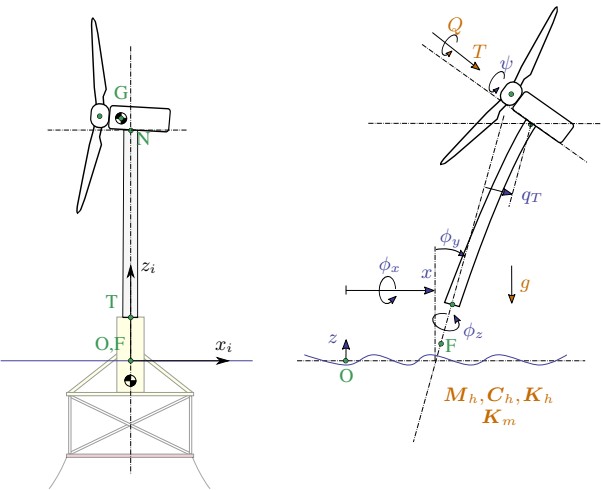

**Figure 3.** Notations for the structural modeling of the floating wind turbine, assuming no yawing of the nacelle. Left: main points $(F, T, N, O, G)$ and inertial coordinate system $(i)$. Right: degrees of freedom $(x, y, z, \phi_x, \phi_y, \phi_z, q_t, \psi)$ and main loads: aerodynamics $(T, Q)$, hydrodynamics ($6 \times 6$ mass and damping and stiffness matrices, $\boldsymbol{M}_h$, $\boldsymbol{C}_h$, $\boldsymbol{K}_h$; wave excitation force neglected), mooring ($6 \times 6$ stiffness matrix, $\boldsymbol{K}_m$), and gravity $(g)$.


rigid body, and its motion is described using 6 DOF: surge, sway, heave, roll, pitch, and yaw, respectively noted $x$, $y$, $z$, $\phi_x$, $\phi_y$,
and $\phi_z$. The tower bending in the fore-aft direction is represented using 1 generalized DOF, $q_t$, associated with a Rayleigh-Ritz
shape function, taken as the first fore-aft mode shape of the tower (Branlard, 2019). The side-side tower bending can be added
in a similar way, but for simplicity, it was not considered in this study. The shape function along the tower height, $z_t$, is written

as $\Phi(z_t)$, with $\Phi(0) = 0$ at the tower bottom, and $\Phi(L_T) = 1$ at the tower top, where $L_T$ is the tower length. The shaft rotation is noted $\psi$, so that the rotation speed of the rotor is $\dot{\psi}$, where the dot notation indicates differentiation with respect to time. The rotor-nacelle assembly is modeled as a rigid body. The full vector of DOF is therefore $\boldsymbol{q} = [x, \ y, \ z, \ \phi_x, \ \phi_y, \ \phi_z, \ q_t, \ \psi]$. The equations of motion will be recast into a first-order form by concatenating the vector of DOF and its time derivative, $\boldsymbol{x} = [\boldsymbol{q}, \dot{\boldsymbol{q}}]$. The selected set of DOF capture the first-order effects as it is the minimal set required to capture the full motion of the floater (necessary to compute restoring loads and tower loads), the tower flexibility (necessary to capture tower loads) and the rotor motion (necessary to capture the aerodynamics). Additional degrees of freedom could be considered to increase the modeling accuracy, in particular to include floater flexibility for internal calculation of substructure member loads. This would increase the computational requirement and only contribute to second-order effects, and it is therefore postponed to future work.

In this work, we perform simplifying assumptions, e.g., neglecting the influence of nacelle yaw on the system. The measurement data are conveniently provided in the fore-aft and side-side system of the nacelle. The main assumption is therefore that we assume a rotational symmetry of the platform and mooring system about the yaw axis. We intend to lift this assumption in future work. Some of the consequences of this assumption is that we do no capture changes of inertial properties due to asymmetry of the support structure and changes of stiffness of the mooring system. In the case of the TetraSpar, the mass matrix of the floater does not vary significantly with the yawing of the coordinate system, and the assumption is fair. We note that if the structure had perfect $120 \deg$ symmetry about the yaw axis, then its inertia would be invariant by yaw rotation. For the restoring stiffness of the mooring system, the diagonal terms do not vary significantly as the coordinate system yaws, but some of the coupling terms vary by 50% to 200%. The couplings between the platform DOF are likely wrongly estimated under the rotational symmetry assumption. The impact is nevertheless limited because most of the platform DOF ($x, y, \phi_x$, and $\phi_y$) are measured and therefore observable by the Kalman filter.

## 3 Individual components of the digital twin

In this section, we describe and verify the individual components of the digital twin presented in Figure 1. In section 4, we present applications of the digital twin where all the individual components are combined together.

### 3.1 Wind turbine measurements

The measurements used as inputs to the digital twin are listed in Table 1. These outputs are stored in a database at a sampling rate of 25 Hz. We expect these measurements to be standard sensors for any floating wind turbine. The TetraSpar prototype is equipped with additional measurements that are used to validate the implementation of the digital twin (see section 4).

**Table 1.** Measurements used as inputs to the digital twin.

| Signal | Symbol |
|---|---|
| Collective blade pitch angle | $\theta_p$ |
| Rotor speed | $\dot{\psi}$ |
| Generator torque* | $Q_g$ |
| Platform surge and sway | $x, y$ |
| Platform roll and pitch | $\phi_x, \phi_y$ |
| Nacelle accelerations | $\ddot{\boldsymbol{r}}_N$ |

\* Obtained from the power measurement using Equation 2.

## 3.2 Nonlinear wind turbine models

### 3.2.1 Overview

Similar to our previous work (Branlard et al., 2020b), we use two different pathways to obtain nonlinear and linear models of floating wind turbines: OpenFAST and WELIB (Wind Energy LIBrary, Branlard (2023b)). The OpenFAST approach was described in subsubsection 2.3.2, it is compared to the WELIB approach in Table 2 and the WELIB toolset is further discussed below. In the next sections, we will show that the results from both approaches are consistent with each other so that either of the two can be used to obtain nonlinear and linear reduced order models. Ultimately, in section 4, a mix of the two approaches is used for the digital twin: linear OpenFAST models for the state-space equations (subsection 3.5) and WELIB for the virtual sensing step (subsection 3.6).

**Table 2.** Approaches and tools used to obtain nonlinear and linear models.

| Approach | Tool | Usage | Formulation & Linearization |
|---|---|---|---|
| **①** OpenFAST | **OpenFAST** (ElastoDyn, HydroDyn, MAP, AeroDyn) | Structural model Hydrodynamics Moorings Aerodynamics Virtual sensing | Numerical and analytical |
| **②** WELIB (Python tools) | **YAMS** | Structural model Virtual sensing | Analytical |
| | **pHydroDyn** | Hydrodynamics | Numerical |
| | **pyMAP** | Moorings | Numerical |

### 3.2.2 WELIB tools

The WELIB approach consists of a set of dedicated open-source Python tools that are similar to the ElastoDyn, HydroDyn and MAP modules of OpenFAST. We developed these tools to offer additional modularity and granularity: the tools can be

called individually or together; their states, inputs and outputs can be accessed and manipulated at each time step; and the Python scripting eases the manipulation of the models. For instance, this allows for: 1) analytical linearization of the structural dynamics, 2) simple linearization of the hydrodynamics (obtention of $6 \times 6$ matrices), 3) linearization of hydrodynamics with respect to wave elevation, 4) linearization with respect to parameters (Jonkman et al., 2022), and 5) interactive time-stepping of the linear and nonlinear model. In this work, we mostly use the first two features listed above and their usage will be described in subsubsection 3.3.2. Results from time-stepping simulations will be presented in subsubsection 3.3.3. We expect to exploit the additional features of WELIB in future digital-twin implementations. For this work, we implemented the following tools in WELIB: 1) YAMS, a symbolic structural dynamics package to obtain the equations of motion of an assembly of rigid and flexible bodies analytically, and allow for their analytical linearization (Branlard and Geisler, 2022); 2) pHydroDyn, a Python version of the module HydroDyn (with a subset of HydroDyn's functionality) to determine the hydrodynamic loads; and 3) pyMAP, a wrapper around the MAP module of OpenFAST, to obtain the mooring quasi-statics. With these three additions, it is possible to perform nonlinear simulations of floating wind turbines using WELIB and perform comparisons with OpenFAST.

### 3.2.3 Differences between the two nonlinear approaches

Currently, no controller or aerodynamic module is present in WELIB. Therefore, nonlinear timestepping simulations with WELIB are limited to free-decay simulations or prescribed loads. Another shortcoming is that WELIB does not cover the full range of options available with OpenFAST, which is a continuously evolving, extensively verified and validated tool. Such options include the potential flow representation of hydrodynamic bodies, the flexibility of the floating structure, aerodynamic and control features. One benefit of WELIB over OpenFAST is the possibility to perform interactive time-stepping, that is, to change the states and inputs dynamically during the simulation. We do not use this approach in this work, but it can be considered for nonlinear digital twin applications, for instance, using an extended Kalman filter algorithm. Another benefit is the possibility to obtain analytical linear models of the structure, which avoids using finite-differences and therefore reduces the associated numerical errors. In the WELIB approach, the individual modules are linearized separately before being combined into the final linear model, and it is therefore easier to understand where each term in the Jacobians of the linear models comes from, and thereby, gain physical intuitiveness on the model. Ultimately, the linear models obtained by both approaches are similar and differ mostly based on differences in the structural dynamics equations and the implementation of rotational transformation matrices. Results comparing time simulations using both approaches will be presented in subsection 3.3.

### 3.3 Linear wind turbine models

As part of our digital twin concept, we have chosen to use linear wind turbine models and a Kalman filter for the core of the state estimation (see subsection 3.5). Nonlinear models and an extended Kalman filter could be considered in future iterations. In this section, we describe how the linear models from OpenFAST and WELIB are obtained.

### 3.3.1 OpenFAST linearization

OpenFAST can provide full-system linearization of its underlying nonlinear models by using a mix of analytically and finite-difference-derived Jacobians (Jonkman and Jonkman, 2016; Jonkman et al., 2018). The linearization process provides the state-space model ($\delta\dot{\boldsymbol{x}} = \boldsymbol{A}\delta\boldsymbol{x} + \boldsymbol{B}\delta\boldsymbol{u}$) and output equation ($\delta\boldsymbol{y} = \boldsymbol{C}\delta\boldsymbol{x} + \boldsymbol{D}\delta\boldsymbol{u}$) for small perturbations (indicated with $\delta$) of the internal states ($\boldsymbol{x}$), inputs ($\boldsymbol{u}$), and outputs ($\boldsymbol{y}$) of OpenFAST, around a selected operating point. OpenFAST provides the linear model for the entire set of states, inputs, and outputs present in the model (including virtual sensor-type outputs typically written to an output file and not used internally). In this work, we extract subsets of the $\boldsymbol{A}$, $\boldsymbol{B}$, $\boldsymbol{C}$, and $\boldsymbol{D}$ matrices and combine them to form the linear model of the state estimator (see subsection 3.5).

### 3.3.2 WELIB linearization

WELIB performs the linearization of the structure, hydrodynamics, and moorings independently before combining them into one model. The aerodynamic loads are not linearized because a dedicated aerodynamic estimator is used in this work (see subsection 3.4). The steps are as follows:

- The structural equations are linearized analytically using our symbolic framework (Branlard and Geisler, 2022). We introduced a notion of "augmented inputs" to linearize the equations of motion without an explicit knowledge of the external forces. The process is described in Appendix A.

- We compute the $6\times6$ linearized rigid-body hydrodynamics matrices (mass matrix $\boldsymbol{M}_h$, damping matrix $\boldsymbol{C}_h$, and stiffness matrix $\boldsymbol{K}_h$) corresponding to the six rigid-body motions of the platform. At the time of this study, these matrices could not be obtained directly from OpenFAST. While working on this issue, we ended up devising multiple ways to obtain them. They can now be obtained using: 1) full-system linearization of the HydroDyn module, 2) the Python implementation of the HydroDyn module by performing rigid-body perturbations of the full platform, or 3) an upgraded version of the OpenFAST HydroDyn driver that also uses rigid-body perturbations. The first approach uses baseline OpenFAST functionalities but requires additional postprocessing scripts and derivations. The full-system linearization of OpenFAST provides Jacobians of the hydrodynamic loads as a function of motions of the individual hydrodynamic analysis nodes (of which models often have hundreds to thousands of). To transfer these individual Jacobians to the reference point and obtain the $6 \times 6$ matrices, we developed and used the method presented in Appendix B. The process is involved and prone to errors. In comparison, the second and third approaches are straightforward to implement and are several orders of magnitude faster. The Python version was implemented first and then ported over to Fortran so that it can be readily available to the OpenFAST community. The consistency between the different approaches was verified, and because of its ease of use, the second approach is retained in this study. We note that in this study, all members are modeled using the Morison equation and the hydrodynamic drag is set to zero during the linearization process. There is therefore no frequency-dependent damping, and the effect of hydrodynamic drag is assumed to be part of the modeling uncertainty of the state estimator (see subsection 3.5).

– The linearized $6 \times 6$ mooring stiffness matrix, $\boldsymbol{K}_m$, is obtained by calling the linearization feature of the MAP module,
and transferring the Jacobian to the reference point using the method outlined in Appendix B.
– The linearized equations of motion are assembled as:
$$[\boldsymbol{M}_0 + \boldsymbol{Q}_0 \boldsymbol{M}_h] \delta \ddot{\boldsymbol{q}} + [\boldsymbol{C}_0 + \boldsymbol{Q}_0 \boldsymbol{C}_h] \delta \dot{\boldsymbol{q}} + [\boldsymbol{K}_0 + \boldsymbol{Q}_0 (\boldsymbol{K}_h + \boldsymbol{K}_m)] \delta \boldsymbol{q} = \delta \boldsymbol{f}_a + \delta \boldsymbol{f}_h \tag{1}$$
where the matrices with subscript 0 originate from the linearization of the structure (see Appendix A). The matrix $\boldsymbol{Q}_0$,
of dimension $8 \times 6$, maps the subset of the 6 rigid-body platform DOF $(x, y, z, \phi_x, \phi_y, \phi_z)$, used for the definitions of
$M_h, C_h, K_h$ and $K_m$, to the full vector of DOF, $\boldsymbol{q}$. The term $\delta \boldsymbol{f}_a$ is an approximation of the aerodynamic loads and will
be discussed in subsection 3.4. The term $\delta \boldsymbol{f}_h$ is an approximation of the hydrodynamic wave-excitation loads. In this
work, $\delta \boldsymbol{f}_h$ is mapped into the inherent model noise of the Kalman filter (see subsection 3.5). Assuming that the loading
is part of the model noise is a crude approximation that is expected to be fair as long as the loading has a zero mean
value, which is expected to be the case for the wave loading, but not for the wind or current loading (here omitted). This
modeling choice is not very influential in this work because the motions of the platform measured by the inclinometers
and GPS sensors inherently carry information about the wave loading. Improvements could be obtained by including
a model for the wave-excitation loads, and further, limiting the wave load signal such that it remains within a certain
frequency band.
For instance, we could introduce a hydrodynamic state analog to the wave elevation or a set of states that scales different
hydrodynamic shape functions so that the hydrodynamic load can be obtained as a linear superposition of scaled shape
functions. In our application (tower section loads), such modeling did not appear necessary, but it will be considered in
future work as it can be relevant to estimate substructure loads.
– We recast Equation 1 into a first-order system to obtain the state matrix $\boldsymbol{A}$.
**3.3.3    Verification of the linear models**
In this section, we compare results from the OpenFAST nonlinear model, the OpenFAST linear model, and the WELIB linear
model for free-decay simulations of the TetraSpar structure. Free-decay simulations are sufficient because wave and aerody-
namic loads are purposely not included in the linear models used by the digital twin. The OpenFAST linear model is obtained
about the operating point defined by $\boldsymbol{q}_0 = \boldsymbol{0}$ and $\dot{\psi}_0 = 10$ rpm. All models (including the OpenFAST nonlinear model) use
8 DOF. The initial conditions are set to $\boldsymbol{q} = [1, -1, 0.6, 0.5, 0.5, 0, -0.2, 0]$ (in m and deg) and $\dot{\psi} = 10$ rpm, after which the
structure is free to move.
First, simulation without hydrodynamics (structure only) are considered, to isolate and verify the structural-dynamics part
of the models. The time responses from the linear and nonlinear models are in strong agreement when only the structure is
considered (see results in Appendix C). Then, we consider results for a model that includes hydrodynamics but without wind or
external waves (still water). We set the hydrodynamic drag to zero due to the difficulty in linearizing this term and let the state
estimator account for this modeling uncertainty. Results of the free-decay simulation are given in Figure 4 for a time period

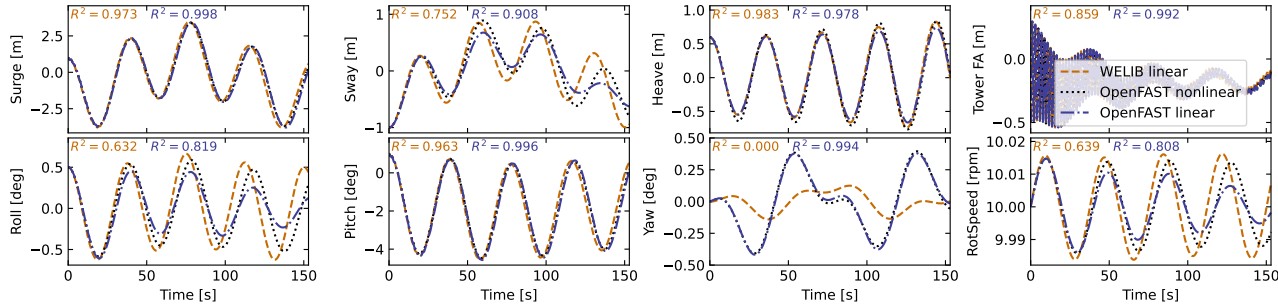

**Figure 4.** Free decay of the structure using nonlinear and linear models for a case including moorings and hydrodynamics (still water). Time series of the main DOF.

of 153 s corresponding to the surge frequency. When hydrodynamics is included, the time responses from the linear models are in strong agreement with the nonlinear OpenFAST results for the surge, heave, pitch, and tower fore-aft DOF. The sway, roll, and rotor speed responses tend to drift as the simulation time advances, which we assume can be attributed to inherent differences between linear and nonlinear models. The coefficient of determination ($R^2$) is indicated in Figure 4, comparing the linear models to the reference OpenFAST simulations for each response. In all cases, the OpenFAST linear model is closer to the nonlinear OpenFAST model than the WELIB model. The consistency between the linear and nonlinear OpenFAST model is expected because they are obtained from the same code base. The WELIB linear model had difficulty capturing the yaw response. We believe that some of the error in the yaw signal is due to differences between the formulations of the three-dimensional rotations in OpenFAST and WELIB. The difference in yaw, results in a difference of coupling between the DOF, which can explain the differences observed in the sway, roll and rotor speed signals.

To further quantify the differences between the models, we compare the natural frequencies obtained using the OpenFAST linear and WELIB linear models in Table 3. Overall, the frequencies between the two linear formulations agree very well (less

**Table 3.** Comparison of system frequencies obtained using the WELIB and OpenFAST linear models with and without hydrodynamics (no added mass, damping, hydrostatics, or wave excitation)

| Mode | Structure + mooring | | | Structure + mooring + hydrodynamics | | |
|---|---|---|---|---|---|---|
| | OpenFAST [Hz] | WELIB [Hz] | Rel. Err [%] | OpenFAST [Hz] | WELIB [Hz] | Rel. Err [%] |
| Surge | 0.0088 | 0.0088 | -0.2 | 0.0067 | 0.0065 | -2.4 |
| Sway | 0.0088 | 0.0088 | -0.1 | 0.0067 | 0.0068 | 0.7 |
| Yaw | 0.0163 | 0.0162 | -1.0 | 0.0128 | 0.0128 | -0.3 |
| Pitch | 0.0879 | 0.0886 | 0.7 | 0.0253 | 0.0257 | 1.6 |
| Roll | 0.0894 | 0.0902 | 0.9 | 0.0256 | 0.0266 | 4.0 |
| Heave | NA | NA | NA | 0.0276 | 0.0276 | -0.2 |
| Tower FA | 0.5782 | 0.5789 | 0.1 | 0.5129 | 0.5145 | 0.3 |

than 2.5% relative error), except for the roll frequencies (4% error) with hydrodynamics. Given the results of this section, we
will continue this study using the OpenFAST linear model. We expect that continuous development of WELIB will further
narrow the gap with OpenFAST in the future.

## 3.4 Aerodynamic estimator

In subsection 3.3, we indicated that the linear models were derived without accounting for aerodynamics. Instead, we choose
to include the aerodynamic contribution separately within the digital twin. The reason for this choice is that the determination
of the aerodynamic loads is essential to capturing the main loading and deflections of the structure, in particular the tower, and
the aerodynamic loads vary significantly over the range of operating conditions. Therefore, separating this contribution limits
the need to obtain different linearized models for different operating conditions. We have successfully applied this approach
in the past (Branlard et al., 2020a). In this work, we extend this approach to accommodate the floating wind application. The
different elements of the aerodynamic estimator consist of a torque estimator, aerodynamic maps, and a wind speed estimator.

### 3.4.1 Kalman filter for torque estimation

We assume that the power and rotor speed are reliable measurement signals, and we further assume that the generator torque
(relative to the low-speed shaft) can be inferred from the power signal as:
$$Q_g = \frac{P}{\dot{\psi}} \frac{1}{n \eta_{\mathrm{DT}}(\dot{\psi})} \tag{2}$$
where $\eta_{\mathrm{DT}}$ is the drivetrain (gearbox and generator) efficiency and $n$ is the gear ratio. For the TetraSpar, $n = 1$, and we assume
$\eta_{\mathrm{DT}} = 1$. The dynamics equation of the drivetrain is modeled as:
$$J_{\mathrm{DT}}\ddot{\psi} = Q - Q_g \tag{3}$$
where $J_{\mathrm{DT}}$ is the inertia of the drivetrain about the shaft axis. If we assume that the generator torque is a measurement, then an
augmented Kalman filter (Lourens et al., 2012) can be used to estimate the aerodynamic torque $Q$, using the following state
equation:
$$\begin{bmatrix} \dot{\psi} \\ \ddot{\psi} \\ \dot{Q} \end{bmatrix} = \begin{bmatrix} 0 & 1 & 0 \\ 0 & 0 & \frac{1}{J_{\mathrm{DT}}} \\ 0 & 0 & 0 \end{bmatrix} \begin{bmatrix} \psi \\ \dot{\psi} \\ Q \end{bmatrix} + \begin{bmatrix} 0 \\ -\frac{1}{J_{\mathrm{DT}}} \\ 0 \end{bmatrix} Q_g \tag{4}$$
A random walk approach is used for the evolution of the torque, that is, $\dot{Q} = 0$, and the Kalman filter adds further model noise
to this equation. The measurement equation of the Kalman filter is:
$$\begin{bmatrix} \dot{\psi} \\ Q_g \end{bmatrix} = \begin{bmatrix} 0 & 1 & 0 \\ 0 & 0 & 0 \end{bmatrix} \begin{bmatrix} \psi \\ \dot{\psi} \\ Q \end{bmatrix} + \begin{bmatrix} 0 \\ 1 \end{bmatrix} Q_g \tag{5}$$
In the following, we write $\hat{Q}$, the aerodynamic torque obtained using the method outlined above. We present verification results
in subsubsection 3.4.4.

### 3.4.2  Aerodynamic maps

It is commonly accepted that the aerodynamic performance of a wind turbine mostly depends on the tip-speed ratio and the
pitch angle of the blade. With compliant structures, the bending of the blade, the bending of the tower, and the motions of the
floating platform (in particular, the platform pitch) will also affect the aerodynamic performance. These motions are to a large
extent a function of the mean wind speed. Therefore, we recommend tabulating the aerodynamic performance as a function
of wind speed ($U$), rotor speed ($\dot{\psi}$), blade pitch ($\theta_p$), and platform pitch ($\phi_y$, assumed to be in the fore-aft direction). The
power and thrust coefficients, respectively noted $C_P$ and $C_T$, are precomputed using aeroelastic simulations in OpenFAST for
a discrete set of values of the four input parameters. In the simulations, the blade and tower elasticity are accounted for. To
limit the number of simulations, only the points that are within reasonable proximity of the regular operating conditions of the
wind turbine are computed. The 4D aerodynamic maps are precomputed as follows:
$$C_P(U,\dot{\psi},\theta_p,\phi_y), \quad C_T(U,\dot{\psi},\theta_p,\phi_y) \tag{6}$$
$$U \in \{2,3,\cdots,25\} \text{ m.s}^{-1}, \quad \dot{\psi} \in \{5,5.5,\cdots,18\} \text{ rpm}, \tag{7}$$
$$\theta_p \in \{-1,0,\cdots,30\} \text{ deg}, \quad \phi_y \in \{-10,0,15\} \text{ deg} \tag{8}$$
The precomputed values are stored in a database.

### 3.4.3  Wind speed estimation

The digital twin uses the aerodynamic map database to estimate the wind speed and aerodynamic thrust. For a given air density
($\rho$), rotor radius ($R$), and measurements $\tilde{\dot{\psi}}$, $\tilde{\theta}_p$, $\tilde{\phi}_y$, the aerodynamic torque and thrust are readily obtained as a function of
wind speed from the database:
$$Q(U) = \frac{1}{2}\rho\frac{U^3}{\tilde{\dot{\psi}}}\pi R^2 C_P(U,\tilde{\dot{\psi}},\tilde{\theta}_p,\tilde{\phi}_y), \qquad T(U) = \frac{1}{2}\rho U^2 \pi R^2 C_T(U,\tilde{\dot{\psi}},\tilde{\theta}_p,\tilde{\phi}_y) \tag{9}$$
where SI units are assumed for all variables. For a given estimated torque ($\hat{Q}$), the estimated wind speed ($\hat{U}$) is found such that:
$$Q(\hat{U}) - \hat{Q} = 0 \tag{10}$$
As illustrated in Figure 5, multiple values of $\hat{U}$ can potentially satisfy Equation 10 because the aerodynamic torque is a
nonlinear function of the wind speed. In such cases, we use the steady-state operating condition curve of the turbine to choose
between the multiple solutions (typically two) by selecting the point closest to this curve (see Figure 5). A relaxation scheme
is also used, based on the previous estimate, to alleviate sudden jumps of the estimated wind speed.

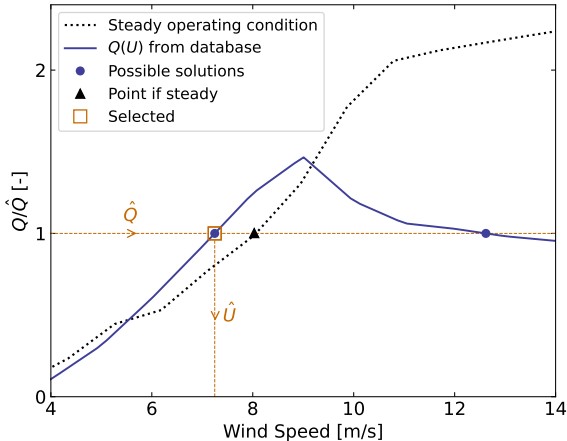

**Figure 5.** Illustration of wind speed estimation in the case where multiple wind speed values match the target torque value $\hat{Q}$

### 3.4.4 Verification of the aerodynamic estimator

To verify the aerodynamic estimator, we ran an OpenFAST simulation of the TetraSpar with the "turbulent step" wind field mentioned in subsubsection 2.3.2 and irregular waves computed with a significant wave height of $H_s = 6$ m and a peak spectral period of $T_p = 14$, which represent a fairly extreme sea state for the site of the TetraSpar prototype. The simulated values of $\dot{\psi}$, $\theta_p$, $\phi_y$, $Q_g$ are used as direct input to the aerodynamic estimator. Comparisons of the estimates with the OpenFAST outputs are shown in Figure 6. The shaded areas on the graphs represent the areas where the generator torque is zero (turbine spinning up);

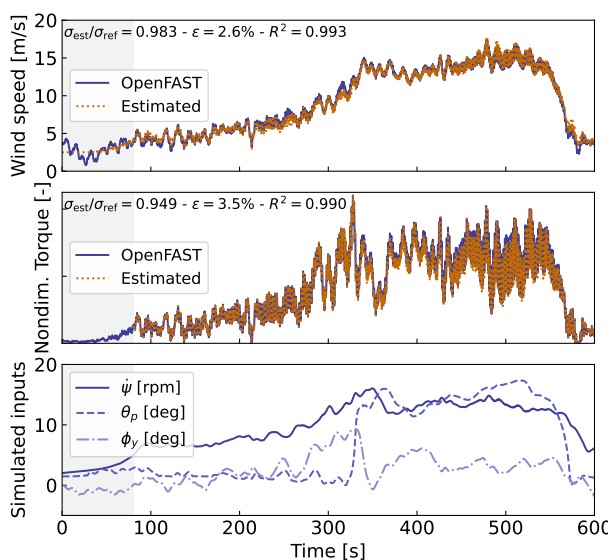

**Figure 6.** Example of aerodynamic estimation using "simulated measurements" from OpenFAST. Top: wind speed. Middle: Dimensionless torque. Bottom: structural inputs from the OpenFAST simulation provided to the estimator.

therefore, the wind speed estimator is not expected to work in that region. The tops of the plots indicate the ratio of standard
deviations, the mean relative error ($\epsilon$), and the coefficient of determination ($R^2$). Throughout this article, we define the mean
relative error of a quantity $x$ as:
$$\epsilon(x) = \text{mean}_i \left[ \frac{|x_{\text{est}}[i] - x_{\text{ref}}[i]|}{\text{mean}(|x_{\text{ref}}|)} \right] \tag{11}$$
where $x_{\text{est}}$ is the estimated signal, $x_{\text{ref}}$ is the reference signal, and $x[i]$ is the value of a signal at time step $i$. Using the mean
of $|x_{\text{ref}}|$ in the denominator avoids issues related to signals crossing 0. It results in lower mean relative error than if the
instantaneous value were used, but the metric is still indicative of how far the two signals are on average.
To quantify the performance of the estimator, we reproduce the simulation above, but add different noise levels to the
measurements to account for measurement errors by the sensors. A Gaussian noise signal of zero mean and standard deviation
$r\sigma$ is added to each input, where $r$ is the noise level and $\sigma$ is the standard deviation of the clean input. The results are shown in

Table 4. As expected, the error in the estimation increases with increasing noise levels. This numerical experiment provides a

**Table 4.** Mean relative error ($\epsilon$) of the wind speed, torque and thrust estimates for increasing noise levels.

| Noise level | 0% | 1% | 5% | 10% | 20% |
|---|---|---|---|---|---|
| Wind Speed | 2.6% | 2.6% | 3.1% | 4.1% | 6.7% |
| Torque | 3.5% | 3.8% | 5.0% | 6.8% | 11.1% |
| Thrust | 4.1% | 5.1% | 5.6% | 7.3% | 11.6% |


rough quantification of the errors that can be expected from the aerodynamic estimator.
**3.5 State estimator**
In this work, we follow a similar approach to our previous work (Branlard et al., 2020a), where an augmented Kalman filter
is used to estimate states and loads. The Kalman filter used in the aerodynamic estimator (subsection 3.4) is augmented with
additional states and outputs. The Kalman filter uses two linear models: a state-equation, describing the time evolution of the
states, and an output equation, describing how the measurements are related to the states and inputs. The state and output
equations are written:
$$\delta\dot{\boldsymbol{x}}_{\text{KF}} = \boldsymbol{X}_x \delta\boldsymbol{x}_{\text{KF}} + \boldsymbol{X}_u \delta\boldsymbol{u}_{\text{KF}} + \boldsymbol{w}_x \tag{12}$$
$$\delta\boldsymbol{y}_{\text{KF}} = \boldsymbol{Y}_x \delta\boldsymbol{x}_{\text{KF}} + \boldsymbol{Y}_u \delta\boldsymbol{u}_{\text{KF}} + \boldsymbol{w}_y \tag{13}$$
where $\delta\boldsymbol{x}_{\text{KF}}$, $\delta\boldsymbol{u}_{\text{KF}}$, and $\delta\boldsymbol{y}_{\text{KF}}$ are the state, input, and output[1], respectively; $\boldsymbol{X}_x$, $\boldsymbol{X}_u$, $\boldsymbol{Y}_x$, and $\boldsymbol{Y}_u$ are the system matrices that
relate the different system vectors; and, $\boldsymbol{w}_x$ and $\boldsymbol{w}_y$ are Gaussian processes represented modeling noise. The output vector,
$\delta\boldsymbol{y}_{\text{KF}}$, is also referred to as the "measurement" vector because it corresponds to the measured signals. At a given time step,

---

[1]In general, the Kalman filter system vectors are different from the ones used for the linearization presented in subsection 3.3, therefore the subscript $KF$ (for Kalman Filter) is added to these vectors.

the Kalman filter algorithm uses the system matrices, a set of measurements, and an a priori knowledge of the model and measurement uncertainties to estimate the state vector (Kalman, 1960; Zarchan and Musoff, 2015).

In this work, we design the state estimator such that the state vector contains the structural degrees of freedom ($\delta q$ and $\delta \dot{q}$) and the aerodynamic torque ($Q$), and the input vector consists of the thrust (obtained with the aerodynamic estimator) and the generator torque (obtained from the power). These design choices were guided by our previous work on the topic. For this choice of state and input variables, we build linear models for the state and output equations. We use the linear models described in subsection 3.3 (the $A$, $B$, $C$, $D$ matrices) to populate the system matrices of the Kalman filter. Additional details on how the relevant Jacobians are extracted are given in subsubsection 3.6.1. Given our choice of system vectors, the state equation is:

$$
\begin{bmatrix} \delta \dot{q} \\ \delta \ddot{q} \\ \dot{Q} \end{bmatrix} = \begin{bmatrix} \mathbf{0} & \mathbf{I} & \mathbf{0} \\ \mathbf{A}_{12} & \mathbf{A}_{22} & \frac{\partial \ddot{q}}{\partial Q} \\ \mathbf{0} & \mathbf{0} & \mathbf{0} \end{bmatrix} \begin{bmatrix} \delta q \\ \delta \dot{q} \\ Q \end{bmatrix} + \begin{bmatrix} \mathbf{0} & \mathbf{0} \\ \frac{\partial \ddot{q}}{\partial Q_g} & \frac{\partial \ddot{q}}{\partial T} \\ 0 & 0 \end{bmatrix} \begin{bmatrix} Q_g \\ T \end{bmatrix} + \boldsymbol{w}_x
\tag{14}
$$

where $\mathbf{A_{12}}$ and $\mathbf{A}_{22}$ are the two lower blocks of the $A$ matrix, and $\mathbf{I}$ is the identity matrix. The Jacobians with respect to the loads are extracted from the $B$ and $D$ matrices. A random walk approach is used for the evolution of the torque $Q$ (that is, we set $\dot{Q} = 0$). The output equation, which effectively relates the measurements to the system states and inputs, is set as:

$$
\begin{bmatrix} \delta \tilde{q} \\ \dot{\psi} \\ \ddot{r}_N \\ Q_g \end{bmatrix} = \begin{bmatrix} \frac{\partial \tilde{q}}{\partial q} & \frac{\partial \tilde{q}}{\partial \dot{q}} & \frac{\partial \tilde{q}}{\partial Q} \\ \mathbf{0} & \tilde{I} & 0 \\ \frac{\partial \ddot{r}_N}{\partial q} & \frac{\partial \ddot{r}_N}{\partial \dot{q}} & \frac{\partial \ddot{r}_N}{\partial Q} \\ \mathbf{0} & \mathbf{0} & \mathbf{0} \end{bmatrix} \begin{bmatrix} \delta q \\ \delta \dot{q} \\ Q \end{bmatrix} + \begin{bmatrix} \mathbf{0} & \mathbf{0} \\ 0 & 0 \\ \frac{\partial \ddot{r}_N}{\partial Q_g} & \frac{\partial \ddot{r}_N}{\partial T} \\ 1 & 0 \end{bmatrix} \begin{bmatrix} Q_g \\ T \end{bmatrix} + \boldsymbol{w}_y
\tag{15}
$$

where $\ddot{r}_N$ is the vector of nacelle accelerations, and $\tilde{q} = \{\delta x, \delta y, \delta \phi_x, \delta \phi_y\}$ is the measurements of surge, sway, roll, and pitch as given in Table 1.

The state and output equations are used as part of a Kalman filter algorithm implemented in WELIB, which continuously takes as input the measurements from the wind turbine (corresponding to the left-hand side of Equation 15). The process and covariance matrices used within the Kalman filter algorithm (determining the values of $\boldsymbol{w}_x$ and $\boldsymbol{w}_y$) are populated based on the estimated standard deviations of the different states and outputs. At each time step, the thrust is estimated using the aerodynamic torque of the previous time step and used as input. The result of the Kalman filter is the estimated states and outputs at each time step. Sample simulation results are provided in section 4.

## 3.6 Virtual sensing

Once the states are estimated by the Kalman filter, the virtual sensing step is used to derive quantities of interest (see Figure 1). In this work, we focus on the estimation of the sectional loads along the tower using a physics-based model. We investigate two methods to obtain these loads.

### 3.6.1 OpenFAST linearization outputs

The first method consists of using the linearization outputs of OpenFAST, namely, using a subset of the equation $\delta\boldsymbol{y} = \boldsymbol{C}\delta\boldsymbol{x} + \boldsymbol{D}\delta\boldsymbol{u}$ (see subsubsection 3.3.1). In general, if a quantity of interest is present in the output vector of OpenFAST, it can be retrieved as follows. If the variable is located at the row index $k$ in the vector $\boldsymbol{y}$, then this variable can be obtained from the states and inputs as:

$$[\boldsymbol{y}]_k = [\delta\boldsymbol{y}]_k + [\boldsymbol{y}_0]_k = [\boldsymbol{C}]_k\delta\boldsymbol{x} + [\boldsymbol{D}]_k\delta\boldsymbol{u} + [\boldsymbol{y}_0]_k \tag{16}$$

where $[\cdot]_k$ indicates that the row $k$ of the matrix or column vector is used. In our case, $[\boldsymbol{y}]_k$ in Equation 16 would be the sectional fore-aft bending moment at the height $z_j$ along the tower, noted $\mathcal{M}_y(z_j)$. The advantages of using this method are multiple: 1) the method is directly applicable to any other outputs computed by OpenFAST, 2) the calculation procedure is linear and therefore computationally efficient, 3) if strain measurements are available at given heights, the rows $[\boldsymbol{C}]_k$ and $[\boldsymbol{D}]_k$ could be included in the output equation of the Kalman filter (Equation 15) to provide information about the model's expectation of these measurements, and 4) the underlying linear model is consistent with the nonlinear model of OpenFAST. The downside of the method is its linearity, in the sense that it is only valid close to the operating point and could lack important nonlinear effects. The values of $[\boldsymbol{C}]_k$, $[\boldsymbol{D}]_k$, and $[\boldsymbol{y}_0]_k$ would potentially need to be reevaluated if the system operates away from the linearized operating point. One possible solution is to introduce gain-scheduling to continuously modify the linear system based on the estimated wind speed. In this work, we used one operating point only and obtained results with fair accuracy (see section 4). We nevertheless expect that to better represent the different operating regions of a pitch-regulated wind turbine, three to five linear models, stitched together through gain-scheduling would be needed.

### 3.6.2 Nonlinear calculation (WELIB)

An alternative method consists of computing the section loads based on first principles using the formulation presented in Branlard (2019). The calculation requires a knowledge of the tower-top loads and the full kinematics of the tower and nacelle (position, velocity, and acceleration). At a given time step, the kinematics are computed based on $\boldsymbol{q}$, $\dot{\boldsymbol{q}}$, and $\ddot{\boldsymbol{q}}$. The tower-top loads are estimated based on the aerodynamic loads and the inertial loads of the rotor-nacelle assembly. We describe the method in more detail in Appendix D. The advantages are that nonlinearities are accounted for and the model is valid irrespective of the operating condition. The downside is that this method does not provide any of the four advantages offered by the OpenFAST linearization method.

### 3.6.3 Verification of the section loads calculation

To verify the calculation of the section loads, we use the same "turbulent step" wind field and irregular sea state that were used in subsubsection 3.4.4. We assume that the time series of $\boldsymbol{q}$, $\dot{\boldsymbol{q}}$ and $\ddot{\boldsymbol{q}}$ are entirely known, extracted from the OpenFAST simulation. These time series are provided to the two section loads algorithms: the WELIB nonlinear algorithm and the OpenFAST linear algorithm.

We run two sets of virtual sensing. In the "ideal" set, the loads at the tower top are extracted from OpenFAST results and
provided to the two virtual sensing algorithms. In this ideal case, the linearized operating points of the OpenFAST linear model
is set as the mean of each of the OpenFAST time series values. Results for the ideal case are illustrated in Figure 7. The

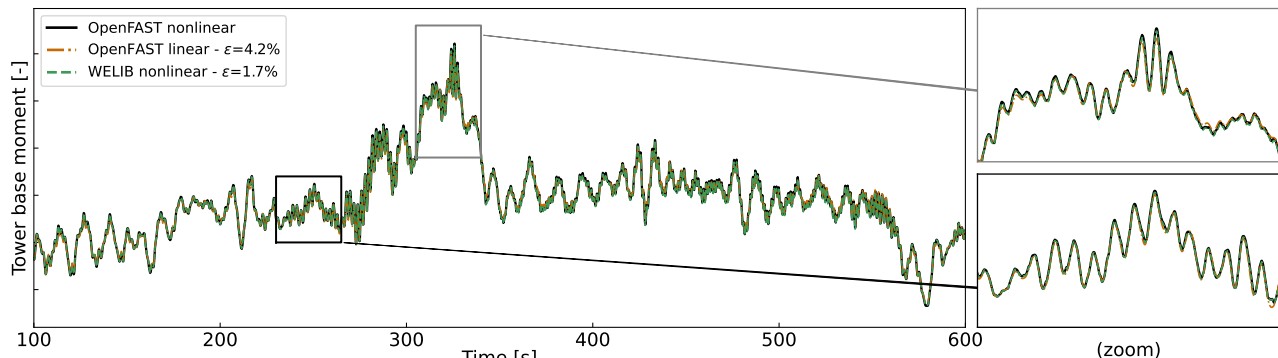

**Figure 7.** Tower fore-aft bending moment for the "turbulent step" and an irregular sea state as calculated by OpenFAST and compared to the WELIB nonlinear and OpenFAST linear method. The motion of the structure is determined by OpenFAST and provided to the two algorithms. The tower-top loads are also provided to the algorithms ("ideal" case, as opposed to Figure 8).


two algorithms are able to reproduce the section loads of OpenFAST with relatively high accuracy, which verifies our two
calculation procedures.
In the second set, labeled "unknown thrust," the tower top loads are not provided to the algorithms; instead, the aerodynamic
estimator mentioned in subsubsection 3.4.4 is used to estimate the aerodynamic loads. This time, we do not set the linearized
operating point of the OpenFAST linear model to the mean value of the time series; we set it to the static equilibrium (without
loading).
The results are illustrated in Figure 8. The accuracy of the section loads calculation is seen to deteriorate when the aerody-
namic loads are estimated with the aerodynamic estimator, which is expected. The damage equivalent load computed with a
Wöhler slope of $m = 5$ is found to be $3.7\%$ lower with the OpenFAST linear method and $1.2\%$ lower with the YAMS nonlinear
method compared to the value for reference signal.
The performance of both algorithms remains satisfactory because the extrapolated signals follow the reference OpenFAST
nonlinear simulation. The relative error obtained with the OpenFAST linear algorithm is higher ($13.3\%$) than the one obtained
using the WELIB nonlinear method ($8.2\%$). The main source of error in the linear model is associated with the fact that the
linearization point was not tuned for this specific simulation. It is our simplifying design choice to use only one linearization
operating point throughout. Because of the loss of accuracy associated with this design choice, we use the WELIB nonlinear
algorithm in the digital twin for the calculation of section loads.
After performing a sensitivity analysis on the inputs and states of the system, we observed that the variables that most affect
the fore-aft section loads are the platform pitch ($\phi_y$), the tower fore-aft bending degree of freedom ($q_t$), and the aerodynamic
thrust. In this section, we assumed that all the states where known (including $\phi_y$ and $q_t$), leading to great accuracy in the

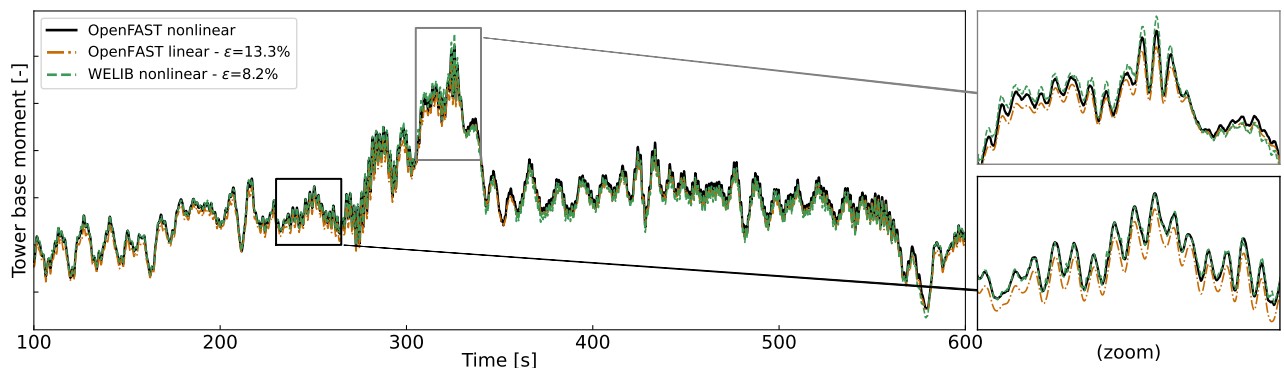

**Figure 8.** Tower fore-aft bending moment for the "turbulent step" and an irregular sea state as calculated by OpenFAST and compared to the WELIB nonlinear and the OpenFAST linear method. The motion of the structure is determined by OpenFAST and provided to the two other algorithms. The tower-top loads are estimated using the aerodynamic estimator ("unknown thrust" case, as opposed to the ideal case presented in Figure 7).

estimation of the section loads. The final verification step involves providing estimated states to the algorithm, which is the topic of the next section.

## 4 Applications of the digital twin

In section 3 we discussed how the different components of the digital twin were introduced and tested using increasing complexity. In this section, we discuss combining the different components to form the digital twin. We begin using numerical experiments from OpenFAST (see subsubsection 2.3.2), similar to what was done previously, before using measurements from the TetraSpar prototype.

### 4.1 Numerical experiment

First, we use the same "turbulent step" wind field and sea state that was used throughout section 3. The augmented states of the system are determined at each time step using the state estimator described in subsection 3.5. The measurements (see Table 1) are taken from the nonlinear OpenFAST simulation. The wind speed and aerodynamic loads are estimated using the aerodynamic estimator described in subsection 3.4. The linear model is derived from linearized OpenFAST, and the section loads in the tower are obtained using the WELIB virtual sensing algorithm described in subsection 3.6. The estimates from the digital twin are compared with the reference nonlinear OpenFAST simulation results in Figure 9. A visual inspection of the time series reveals that the digital twin is able to capture the main trends and fluctuations of the different signals. The match can be considered remarkable given that only the sensors provided in Table 1 are used by the digital twin. Metrics such as mean relative error ($\epsilon$) and coefficient of determination ($R^2$) are indicated on the figure. Despite the visually appealing match, the metrics indicate that the tower-bottom moment has a mean error of $\epsilon = 21\%$. The damage equivalent load of the tower-bottom

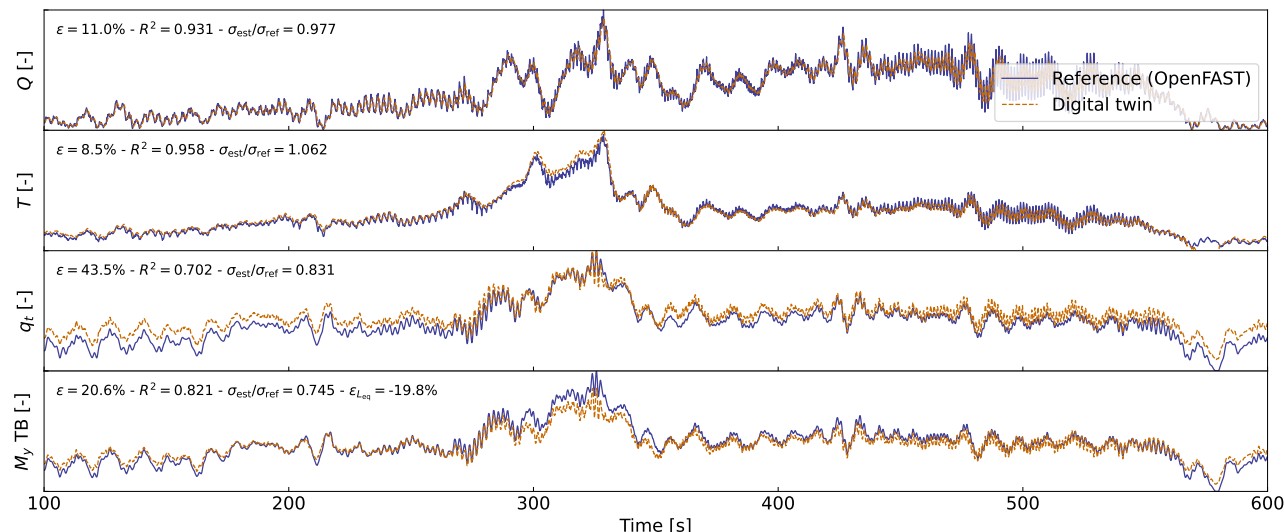

**Figure 9.** Estimated signals from the digital twin compared to results from a nonlinear OpenFAST simulation using the turbulent-step numerical experiment. From top to bottom: aerodynamic torque ($Q$), aerodynamic thrust ($T$), tower-top position ($q_t$), tower-bottom fore-aft bending moment ($M_y$, TB). Results are made dimensionless for confidentiality reasons.

moment is underestimated by $\tilde{\epsilon}_{L_{eq}} = -21\%$, where we define:
$$\tilde{\epsilon}(L_{eq}) = \frac{L_{eq,\text{est}} - L_{eq,\text{ref}}}{L_{eq,\text{ref}}}$$  (17)
Differences in damage equivalent loads typically indicate differences in the frequency content of the signals. We compare the frequency content of the estimated signals with the reference signals in Figure 10. The low-frequency content (below 1 Hz)

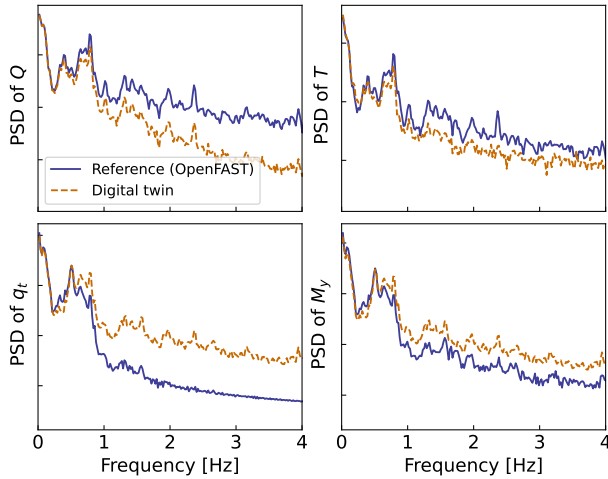

**Figure 10.** Power spectral density (PSD) of the time series presented in Figure 9. A logarithmic scale is used on the $y$ axis.

is captured well, in line with the visual inspection of Figure 9. Unfortunately, no clear trend is found for the high-frequency
content: the power spectra of the aerodynamic loads indicate an underestimation, whereas the spectra of the tower-top position
and tower-bottom bending moment tend to have higher energy content. As shown in previous studies (Branlard et al., 2020a),
filtering of the input measurements can be used to tune the energy content at high frequencies. The method is yet unsatisfactory
because it acts as an artificial rebalancing of energy content to achieve the desired DEL value. Both low and high frequency
content contribute to the DEL values, therefore, we believe that systematic improvement is only possible through modeling
improvements and higher observability of the states by the Kalman filter.
To quantify the errors in the estimation under a wider set of operating conditions, we run 10-min simulations for a set of
wind speeds under normal turbulent conditions and sea states. We select wind speeds from 5 to 20 m/s using 10 different seeds
per bin of wind speed. The seeds are used to randomize the turbulent field and sea states. The wind speed range is selected so
as to avoid cut-in and cut-out events where the aerodynamic estimator is not expected to perform well. The turbulence intensity
is selected based on the normal turbulence model for a turbine of class "A." The wave height and wave period are set as a
function of the wind speed as: $H_s(U) = 0.16U + 1$ and $T_p(U) = 0.09U + 5.57$. The $H_s$ and $T_p$ relationships were obtained by
performing a linear regression on the sea state and wind measurements at the test site. OpenFAST simulations are run for each
case, and then the digital twin is run using these numerical measurements. A summary of the mean relative error on some key

estimated quantities is given in Figure 11. We observe that the mean relative error of the wind speed and aerodynamic loads

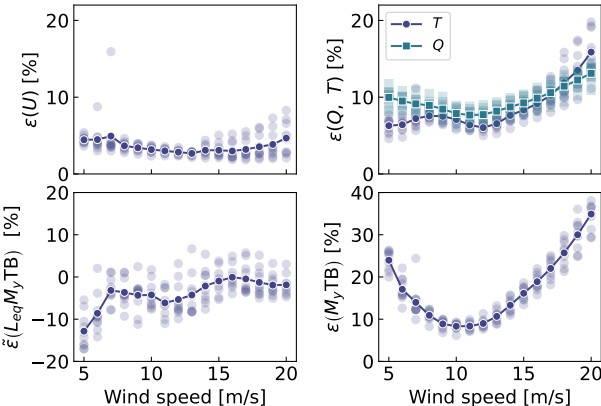

**Figure 11.** Mean relative error of estimated signals for various wind speed and seeds. Clockwise from top left: wind speed $(U)$, aerodynamic loads, tower-bottom moment $(M_y$ TB), and damage equivalent load of the tower-bottom moment $(L_{eq,M_y}$ TB). The individual simulations are indicated by transparent markers. The average over each seed is indicated using solid lines.

is between 5% and 15% with a tendency for larger errors on the aerodynamic loads at low and high wind speeds. The error
further propagates within the system, and the tower-bottom moment is estimated with a relative error between 10% and 40%.
The error levels indicate that the aerodynamic estimator, which is based on quasi-steady rotor-averaged aerodynamics, cannot
fully capture the dynamic aerodynamic state of the rotor in floating conditions. In general, the digital twin lacks sufficient

information to fully capture the tower-top loads and the frequency content of the system. It is expected that the placement of additional sensors, such as accelerometers or load cells, along the tower can significantly improve the estimation of the tower loads (in that case, we would either use OpenFAST linearization outputs or an extended Kalman filter and a nonlinear model for the outputs). As seen in Figure 11, the relative error levels on the damage equivalent loads are between $-10\%$ and $5\%$, with the loads being either overestimated or underestimated depending on the wind speed. The structural health monitoring system could potentially use the estimated error levels indicated in Figure 11 to provide a confidence interval on the fatigue lifetime of the tower. We note that these error levels represent a best-case scenario because we assumed that no noise or biases were present in the measurements. We expect the error levels to increase with additional measurement noise.

## 4.2 Estimations using measurements from the full-scale prototype

In this section, we use measurements from the full-scale TetraSpar prototype installed off the Norwegian coast. Four days of data were selected based on data availability; a wide range of wind speeds are present in the time series (ranging from 4 to 24.5 m/s with an overall mean of 8.9 m/s). Two days were selected in summer and two in winter to account for potential seasonality. Apart from these criteria, the selection of time series can be considered random. The measurement data are stored as 10-min time series sampled at 25 Hz. The total number of 10-min samples used over the four days is 576. The measurement data are provided to the digital twin to perform the state estimation and virtual sensing. The prototype is equipped with load cells at the tower top, middle, and bottom and nacelle wind speed measurements. We use these measurements to compare with the digital twin estimates.

We begin by highlighting the computational time of the current procedure, as computational efficiency is crucial to achieve our digital twin vision. The state estimation is currently 10 times faster than real time. The virtual sensing step is twice as fast as real time, but computational improvements are possible, in particular, by using a compiled language instead of Python. For reference, OpenFAST simulations of the full TetraSpar model (with substructure flexibility) typically run 3 times slower than real time, and a reduced-order OpenFAST model with 8 DOF runs 1.1 times slower. Currently, real time estimation cannot be achieved with OpenFAST. Reduced-order modeling techniques, such as the ones presented in this article, are necessary to implement an online digital twin. Yet, if the digital twin is run as a postprocessing step, then parallelization using multiple CPUs could be used, e.g., processing different time periods of the day.

A sample of results is provided in Figure 12. The figure illustrates a selected case where the estimation of the tower load is reasonably accurate, with an error on the damage equivalent load of only $0.4\%$. We note that the wind speed from the measurement is a point measurement (from the nacelle anenometer, in the wake of the turbine, and moving with the nacelle), and it is therefore not expected to be in strong agreement with the digital twin estimate, which is representative of a rotor-averaged wind speed.

An aggregate of results from all the 10-min digital twin runs is illustrated in Figure 13. The figure shows relative errors in wind speed, thrust, and damage equivalent loads at the tower bottom and tower middle. As indicated previously, the wind speed from the digital twin and the measurements are different quantities, but the level of error obtained indicates that the digital twin is able to capture the main level of wind speed. The aerodynamic thrust from the aerodynamic estimator is compared with

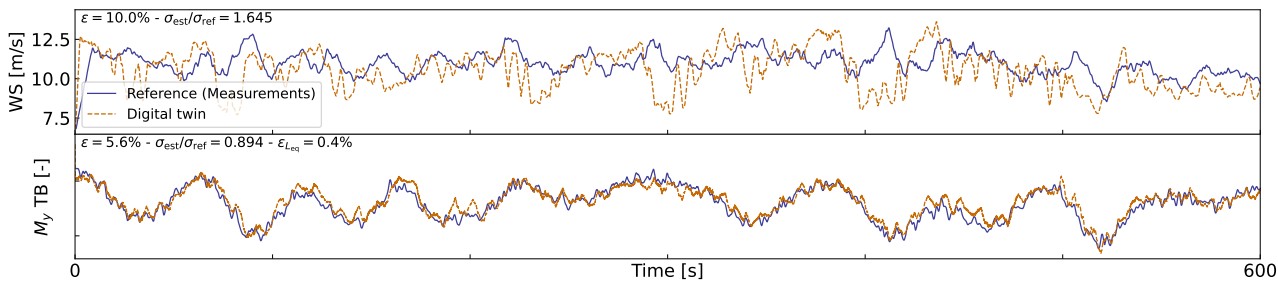

**Figure 12.** Comparison of digital twin outputs with wind speed and tower-bottom moment measurements from the TetraSpar prototype. The measured wind speed comes from a nacelle anenometer and therefore is expected to differ from the rotor-averaged value estimated by the digital twin.

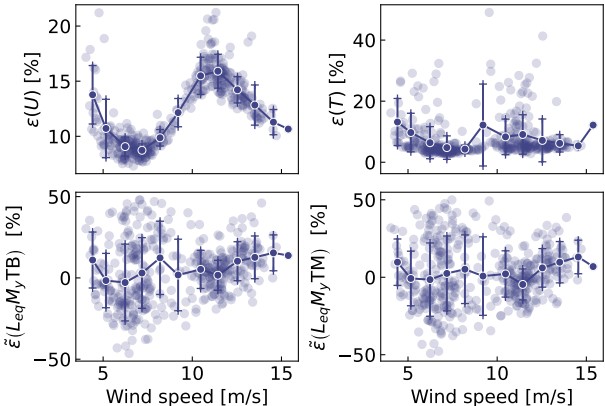

**Figure 13.** Similar to Figure 11 but using measurements from the Tetra Spar prototype. Each marker indicates a 10-min simulation result. Solid lines are bin averages. The whiskers indicate the standard deviation in each bin. The bottom plots are for the tower bottom (TB) and tower middle (TM) bending moments.

the load cell at the tower top in the fore-aft direction. This is a crude first-order approximation (e.g., neglecting inertial and
gravitational loads, nacelle tilting and shaft bending), but the overall estimated levels appears to be, on average, around 10%
from the measured ones. The tower damage equivalent loads are, on average, within $\pm 10\%$ of the values obtained from the
measurements, but some cases show errors ranging between $\pm 50\%$. To give perspective on the large error values taken by the
metrics, we illustrate two cases with large errors in Figure 14 and Figure 15. In both cases, we observe that the estimator is
capturing the trends and low frequencies with accuracy that, from a pure qualitative perspective, would appear satisfactory.
As seen in Figure 14, an offset is present in the signal, which indicates that some physics might be missing from the load
virtual sensing, or that the state estimator is failing. Measurement errors could also affect the results, but no systematic error
was detected over the time period investigated. It is therefore difficult to conclude as to what is the main source of error. In
Figure 14, the overall load level is captured well, but the error in the damage equivalent load is $\epsilon_{L_{eq}}$ is 33%. As illustrated
in Figure 10, our current method fails at capturing the high-frequency content of the signals, which can have a significant

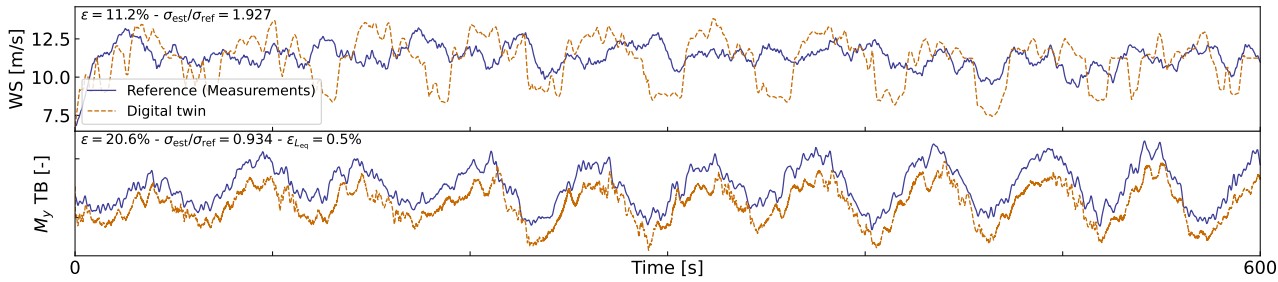

**Figure 14.** Similar to Figure 12, but for a case where a clear offset is present in the tower loads.

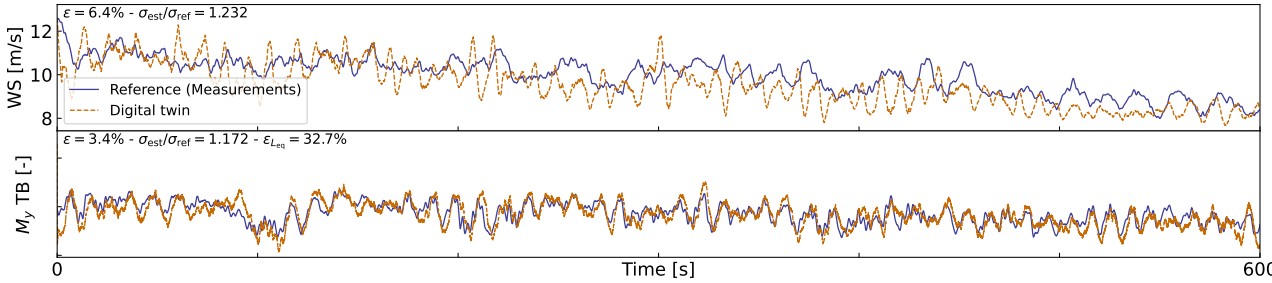

**Figure 15.** Similar to Figure 12, but for a case where a large error in damage equivalent load is observed.

impact on the accuracy of the damage equivalent loads. Despite these challenges, the average accuracy of 10% is promising and indicates that the current methodology can be used to reconstruct some structural and environmental signals from a limited number of readily available sensors.

## 5   Conclusions

In this work, we implemented, verified, and validated a physics-based digital twin solution applied to a floating offshore wind turbine. The work focused on the estimation of the aerodynamic loads and the section loads along the tower, using a set of measurements that we expect to be available on any existing wind turbine (power, pitch, rotor speed, and tower acceleration) and motion sensors that are likely to be standard measurements for a floating platform (inclination and GPS sensors). The key concept behind our approach uses 1) a Kalman filter to estimate the structural states based on a linear model of the structure and measurements from the turbine, 2) an aerodynamic estimator, and 3) a physics-based virtual sensing procedure to obtain the loads along the tower. An important part of the work was developing the methodology and implementing the tools and models necessary for the aerodynamic estimation, state estimation, and load virtual sensing. We explored two different pathways to obtain models: a suite of Python tools and OpenFAST linearization. We used components from both approaches for the digital twin.

Using numerical experiments, we found that the accuracy of the individual models was typically on the order of 5%. When
comparing the digital twin estimations with the measurements from the TetraSpar prototype, the errors increased to 10%–15%
on average for the quantities of interest. Overall, the accuracy of the results appeared promising given the scope of our work,
which aimed to illustrate a proof of concept for a floating wind turbine digital twin. We observed a non-negligible scatter of
results for the estimation of the tower damage equivalent loads that we attributed to the difficulty of capturing high-frequency
content.
Future work should therefore explore possible improvements of the method to address this issue.
Additional improvements could include: 1) gain-scheduling of the linear models to extend the domain of validity of the
linear models used and reduce the modeling error, 2) using nonlinear models and extended Kalman filtering techniques to lift
the linear assumptions that challenges the aerodynamics, hydrodynamics and structural dynamics, 3) introducing additional
degrees of freedom and a full account of the yawing of the nacelle to increase the fidelity of the models and account for the
flexibility of the floater, 4) adding a model to account for wave excitation forces to account for hydrodynamic loads and likely
improve the estimation of member-level loads, 5) introducing additional measurements to improve the state estimation and
increase the observability of the state, 6) improving the robustness of the aerodynamic estimator in particular, beyond the cut-
in and cut-out wind speeds, to apply the digital twin when the turbine is not operating, and, 7) expanding the virtual sensing
steps to estimate additional signals.
*Author contributions.* EB implemented the digital twin and wrote the main corpus of this paper. JJ, CB and JZ provided continuous feedback
on the project and reviewed the article.
*Competing interests.* No competing interests are present.
*Code availability.* The source code of the digital twin and examples using a generic spar turbine are provided in the GitHub repository (Bran-
lard, 2023a).
*Acknowledgements.* This work was authored in part by the National Renewable Energy Laboratory, operated by Alliance for Sustainable
Energy, LLC, for the U.S. Department of Energy (DOE) under Contract No. DE-AC36-08GO28308. Funding provided by U.S. Department
of Energy Office of Energy Efficiency and Renewable Energy Wind Energy Technologies Office. The views expressed in the article do
not necessarily represent the views of the DOE or the U.S. Government. The U.S. Government retains and the publisher, by accepting the
article for publication, acknowledges that the U.S. Government retains a nonexclusive, paid-up, irrevocable, worldwide license to publish or
reproduce the published form of this work, or allow others to do so, for U.S. Government purposes.
*Financial support.* This work was funded under the Technology Commercialization Fund Project, supported by DOE's Wind Energy Tech-
nologies Office.

## Appendix A: Linearization of the equations of motion with augmented inputs

In this section, we describe the procedure used to linearize the structural equations of motion without knowledge of the external loads, which
is used to obtain Equation 1. We write the implicit form of the equations of motion as
$$\mathbf{e}(\boldsymbol{q}, \dot{\boldsymbol{q}}, \ddot{\boldsymbol{q}}, \tilde{\boldsymbol{u}}, t) = \mathbf{0} \tag{A1}$$
where $\boldsymbol{q}$, $\dot{\boldsymbol{q}}$, $\ddot{\boldsymbol{q}}$ and $\tilde{\boldsymbol{u}}$ are the degrees of freedom, velocities, accelerations, and "augmented inputs" of the model, respectively. The term
augmented input is used because the external loads are included in this vector. The external loads are (in general) a function of the degrees
of freedom. Therefore, we write the augmented input vector as:
$$\tilde{\boldsymbol{u}} = \tilde{\boldsymbol{u}}(\boldsymbol{q}, \dot{\boldsymbol{q}}, \ddot{\boldsymbol{q}}, \boldsymbol{u}) \tag{A2}$$
where $\boldsymbol{u}$ is the vector of inputs in the classical sense, that is, consisting of system inputs that do not depend on the degrees of freedom (for
instance, the wave elevation). The operating point is written using the subscript "0," and is defined as:
$$\mathbf{e}(\boldsymbol{q}_0, \dot{\boldsymbol{q}}_0, \ddot{\boldsymbol{q}}_0, \tilde{\boldsymbol{u}}_0, t) = \mathbf{0} \tag{A3}$$
We perturb each variable, as $\boldsymbol{q} = \boldsymbol{q}_0 + \delta\boldsymbol{q}$, $\dot{\boldsymbol{q}} = \dot{\boldsymbol{q}}_0 + \delta\dot{\boldsymbol{q}}$, etc., where $\delta$ indicates a small perturbation of the quantities. The perturbation of the
augmented input is then:
$$\tilde{\boldsymbol{u}} = \tilde{\boldsymbol{u}}(\boldsymbol{q}_0, \dot{\boldsymbol{q}}_0, \ddot{\boldsymbol{q}}_0, \boldsymbol{u}_0) + \left.\frac{\partial \tilde{\boldsymbol{u}}}{\partial \boldsymbol{q}}\right|_0 \delta\boldsymbol{q} + \left.\frac{\partial \tilde{\boldsymbol{u}}}{\partial \dot{\boldsymbol{q}}}\right|_0 \delta\dot{\boldsymbol{q}} + \left.\frac{\partial \tilde{\boldsymbol{u}}}{\partial \ddot{\boldsymbol{q}}}\right|_0 \delta\ddot{\boldsymbol{q}} + \left.\frac{\partial \tilde{\boldsymbol{u}}}{\partial \boldsymbol{u}}\right|_0 \delta\boldsymbol{u} \tag{A4}$$
where $|_0$ indicates that the expressions are evaluated at the operating point. The linearized equations are obtained using a Taylor-series
expansion:
$$\left[\boldsymbol{M}_0 - \boldsymbol{Q}_0 \left.\frac{\partial \tilde{\boldsymbol{u}}}{\partial \ddot{\boldsymbol{q}}}\right|_0\right] \delta\ddot{\boldsymbol{q}} + \left[\boldsymbol{C}_0 - \boldsymbol{Q}_0 \left.\frac{\partial \tilde{\boldsymbol{u}}}{\partial \dot{\boldsymbol{q}}}\right|_0\right] \delta\dot{\boldsymbol{q}} + \left[\boldsymbol{K}_0 - \boldsymbol{Q}_0 \left.\frac{\partial \tilde{\boldsymbol{u}}}{\partial \boldsymbol{q}}\right|_0\right] \delta\boldsymbol{q} = \boldsymbol{Q}_0 \left.\frac{\partial \tilde{\boldsymbol{u}}}{\partial \boldsymbol{u}}\right|_0 \delta\boldsymbol{u} \tag{A5}$$
with
$$\boldsymbol{M}_0 = -\left.\frac{\partial \mathbf{e}}{\partial \ddot{\boldsymbol{q}}}\right|_0, \quad \boldsymbol{C}_0 = -\left.\frac{\partial \mathbf{e}}{\partial \dot{\boldsymbol{q}}}\right|_0, \quad \boldsymbol{K}_0 = -\left.\frac{\partial \mathbf{e}}{\partial \boldsymbol{q}}\right|_0, \quad \boldsymbol{Q}_0 = \left.\frac{\partial \mathbf{e}}{\partial \boldsymbol{u}}\right|_0 \tag{A6}$$
and where $\boldsymbol{M}_0, \boldsymbol{C}_0, \boldsymbol{K}_0$ are the linear mass, damping, and stiffness matrices, and $\boldsymbol{Q}_0$ is the linear forcing vector, also called the input matrix.

## Appendix B: Transfer of a Jacobian from one destination point to another

The Jacobians provided by OpenFAST and MAP are provided at given nodes of the structure (e.g., the hydrodynamic nodes, or the fairleads).
In this section, we highlight the procedure to transfer these Jacobians to another node (the platform reference point) assuming a rigid-body
relationship between the nodes. The procedure is used in this work to compute the linear $6 \times 6$ matrix for the hydrodynamics and mooring
dynamics in subsubsection 3.3.2. We obtain different relationships depending on whether the destination point is assumed to be displaced or
not (see different subsections below).

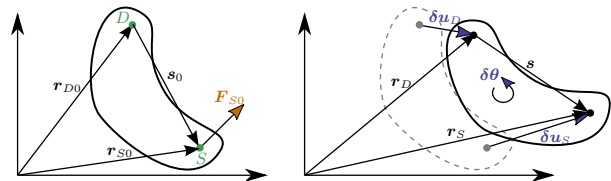

**Figure B1.** Rigid-body kinematics with the loads from one source point ($S$) transferred to ta destination point ($D$), assuming small motion of the points.

## B1 Transfer of Jacobians between two points

We consider a point source (noted $S$) and a destination point (noted $D$). The notations are illustrated in Figure B1. We assume that the two points belong to a rigid body. The forces and moments at the destination and source are related as follows:

$$\boldsymbol{F}_D = \boldsymbol{F}_S \tag{B1}$$

$$\boldsymbol{M}_D = \boldsymbol{M}_S + \tilde{\boldsymbol{s}}\boldsymbol{F}_S \tag{B2}$$

where $\boldsymbol{s} = \boldsymbol{r}_S - \boldsymbol{r}_D$ is the vector from destination point to the source point, $\boldsymbol{F}_S$ and $\boldsymbol{M}_S$ are the force and moments, respectively, at point $S$, and the tilde notation refers to the skew symmetric matrix, which is a matrix representation of the cross product. We seek to linearize Equation B1 and Equation B2 for small displacements and rotations of the destination and source nodes. In particular, we seek to express the Jacobians at the destination node as a function of the source node, assuming a rigid-body relationship between the two. The rigid-body relationship linking the small displacements ($\delta\boldsymbol{u}$) and small rotations ($\delta\boldsymbol{\theta}$) of the source and destination points is:

$$\delta\boldsymbol{u}_D = \delta\boldsymbol{u}_S + \tilde{\boldsymbol{s}}_0\delta\boldsymbol{\theta}_S$$

$$\delta\boldsymbol{\theta}_D = \delta\boldsymbol{\theta}_S \tag{B3}$$

where $\boldsymbol{s}_0$ is the vector between the source and destination points at the operating point (prior to the perturbation). The Jacobians of the transformations given in Equation B3, and its inverse, are:

$$\begin{bmatrix} \frac{\partial \boldsymbol{u}_D}{\partial \boldsymbol{u}_S} & \frac{\partial \boldsymbol{u}_D}{\partial \boldsymbol{\theta}_S} \\ \frac{\partial \boldsymbol{\theta}_D}{\partial \boldsymbol{u}_S} & \frac{\partial \boldsymbol{\theta}_D}{\partial \boldsymbol{\theta}_S} \end{bmatrix} = \begin{bmatrix} \boldsymbol{I} & \tilde{\boldsymbol{s}}_0 \\ \boldsymbol{0} & \boldsymbol{I} \end{bmatrix}, \qquad \begin{bmatrix} \frac{\partial \boldsymbol{u}_S}{\partial \boldsymbol{u}_D} & \frac{\partial \boldsymbol{u}_S}{\partial \boldsymbol{\theta}_D} \\ \frac{\partial \boldsymbol{\theta}_S}{\partial \boldsymbol{u}_D} & \frac{\partial \boldsymbol{\theta}_S}{\partial \boldsymbol{\theta}_D} \end{bmatrix} = \begin{bmatrix} \boldsymbol{I} & -\tilde{\boldsymbol{s}}_0 \\ \boldsymbol{0} & \boldsymbol{I} \end{bmatrix} \tag{B4}$$

To linearize Equation B1 and Equation B2, we introduce the following perturbations:

$$\boldsymbol{F}_D = \boldsymbol{F}_{D0} + \delta\boldsymbol{F}_D, \quad \boldsymbol{F}_S = \boldsymbol{F}_{S0} + \delta\boldsymbol{F}_S \tag{B5}$$

$$\boldsymbol{M}_D = \boldsymbol{M}_{D0} + \delta\boldsymbol{M}_D, \quad \boldsymbol{M}_S = \boldsymbol{M}_{S0} + \delta\boldsymbol{M}_S, \tag{B6}$$

where the subscript 0 indicates values at the operating point. At the operating point, Equation B1 and Equation B2 are satisfied, that is:

$$\boldsymbol{F}_{D0} = \boldsymbol{F}_{S0} \tag{B7}$$

$$\boldsymbol{M}_{D0} = \boldsymbol{M}_{S0} + \tilde{\boldsymbol{s}}_0\boldsymbol{F}_{S0} \tag{B8}$$

## Transfer of forces


Inserting Equation B5 into Equation B1 leads to:


$$\boldsymbol{F}_{D0} + \delta\boldsymbol{F}_D = \boldsymbol{F}_{S0} + \delta\boldsymbol{F}_S \tag{B9}$$


which, using Equation B7, leads to:


$$\delta\boldsymbol{F}_D = \delta\boldsymbol{F}_S \tag{B10}$$


The Jacobians of the loads at node $D$ with respect to the displacements at node $D$ are then obtained by applying the chain rule to Equation B10


and making use of the Jacobian of the displacements given on the right of Equation B4. For instance, for the force:


$$\frac{\partial\boldsymbol{F}_D}{\partial\boldsymbol{u}_D} = \frac{\partial\boldsymbol{F}_S}{\partial\boldsymbol{u}_S}\frac{\partial\boldsymbol{u}_S}{\partial\boldsymbol{u}_D} + \frac{\partial\boldsymbol{F}_S}{\partial\boldsymbol{\theta}_S}\frac{\partial\boldsymbol{\theta}_S}{\partial\boldsymbol{u}_D} = \frac{\partial\boldsymbol{F}_S}{\partial\boldsymbol{u}_S}$$


$$\frac{\partial\boldsymbol{F}_D}{\partial\boldsymbol{\theta}_D} = \frac{\partial\boldsymbol{F}_S}{\partial\boldsymbol{u}_S}\frac{\partial\boldsymbol{u}_S}{\partial\boldsymbol{\theta}_D} + \frac{\partial\boldsymbol{F}_S}{\partial\boldsymbol{\theta}_S}\frac{\partial\boldsymbol{\theta}_S}{\partial\boldsymbol{\theta}_D} = \frac{\partial\boldsymbol{F}_S}{\partial\boldsymbol{\theta}_S} - \frac{\partial\boldsymbol{F}_S}{\partial\boldsymbol{u}_S}\tilde{\boldsymbol{s}}_0 \tag{B11}$$


For the transfer of the moments, the relationship will be different whether the moments are transferred at the undisplaced destination point


or the displaced destination point.


## Moments at the undisplaced destination point


In this section, the moments are transferred to the undisplaced destination point. The vector from the undisplaced destination point to the


displaced source is:


$$\boldsymbol{r} = \boldsymbol{s}_0 + \delta\boldsymbol{u}_S \tag{B12}$$


Introducing Equation B6 and Equation B12 into Equation B2, and temporarily using the "$\times$" notation instead of the tilde notation:


$$\boldsymbol{M}_{D0} + \delta\boldsymbol{M}_D = \boldsymbol{M}_{S0} + \delta\boldsymbol{M}_S + \boldsymbol{s}_0 \times \boldsymbol{F}_{S0} + \boldsymbol{s}_0 \times \delta\boldsymbol{F}_S + \delta\boldsymbol{u}_S \times \boldsymbol{F}_{S0} + \delta\boldsymbol{u}_S \times \delta\boldsymbol{F}_S \tag{B13}$$


Making use of Equation B8, neglecting the nonlinear term ($\delta\boldsymbol{u}_S \times \delta\boldsymbol{F}_S$) and reintroducing the tilde notation leads to:


$$\delta\boldsymbol{M}_D = \delta\boldsymbol{M}_S + \tilde{\boldsymbol{s}}_0\delta\boldsymbol{F}_S - \tilde{\boldsymbol{F}}_{S0}\delta\boldsymbol{u}_S \tag{B14}$$


The Jacobians of the moments at the undisplaced node $D$ with respect to the displacements at node $D$ are then obtained by applying the


chain rule to Equation B14:


$$\frac{\partial\boldsymbol{M}_D}{\partial\boldsymbol{u}_D} = \frac{\partial\boldsymbol{M}_S}{\partial\boldsymbol{u}_S}\frac{\partial\boldsymbol{u}_S}{\partial\boldsymbol{u}_D} + \frac{\partial\boldsymbol{M}_S}{\partial\boldsymbol{\theta}_S}\frac{\partial\boldsymbol{\theta}_S}{\partial\boldsymbol{u}_D} + \tilde{\boldsymbol{s}}_0\left[\frac{\partial\boldsymbol{F}_S}{\partial\boldsymbol{u}_S}\frac{\partial\boldsymbol{u}_S}{\partial\boldsymbol{u}_D} + \frac{\partial\boldsymbol{F}_S}{\partial\boldsymbol{\theta}_S}\frac{\partial\boldsymbol{\theta}_S}{\partial\boldsymbol{u}_D}\right] - \tilde{\boldsymbol{F}}_{S0}\frac{\partial\boldsymbol{u}_S}{\partial\boldsymbol{u}_D}$$


$$= \frac{\partial\boldsymbol{M}_S}{\partial\boldsymbol{u}_S} + \tilde{\boldsymbol{s}}_0\frac{\partial\boldsymbol{F}_S}{\partial\boldsymbol{u}_S} - \tilde{\boldsymbol{F}}_{S0} \tag{B15}$$


and


$$\frac{\partial\boldsymbol{M}_D}{\partial\boldsymbol{\theta}_D} = \frac{\partial\boldsymbol{M}_S}{\partial\boldsymbol{\theta}_S}\frac{\partial\boldsymbol{\theta}_S}{\partial\boldsymbol{\theta}_D} + \frac{\partial\boldsymbol{M}_S}{\partial\boldsymbol{u}_S}\frac{\partial\boldsymbol{u}_S}{\partial\boldsymbol{\theta}_D} + \tilde{\boldsymbol{s}}_0\left[\frac{\partial\boldsymbol{F}_S}{\partial\boldsymbol{\theta}_S}\frac{\partial\boldsymbol{\theta}_S}{\partial\boldsymbol{\theta}_D} + \frac{\partial\boldsymbol{F}_S}{\partial\boldsymbol{u}_S}\frac{\partial\boldsymbol{u}_S}{\partial\boldsymbol{\theta}_D}\right] - \tilde{\boldsymbol{F}}_{S0}\frac{\partial\boldsymbol{u}_S}{\partial\boldsymbol{\theta}_D}$$


$$= \frac{\partial\boldsymbol{M}_S}{\partial\boldsymbol{\theta}_S} - \frac{\partial\boldsymbol{M}_S}{\partial\boldsymbol{u}_S}\tilde{\boldsymbol{s}}_0 + \tilde{\boldsymbol{s}}_0\frac{\partial\boldsymbol{F}_S}{\partial\boldsymbol{\theta}_S} - \tilde{\boldsymbol{s}}_0\frac{\partial\boldsymbol{F}_S}{\partial\boldsymbol{u}_S}\tilde{\boldsymbol{s}}_0 + \tilde{\boldsymbol{F}}_{S0}\tilde{\boldsymbol{s}}_0 \tag{B16}$$


**Jacobian relationships at the undisplaced destination point**

Equation B11, Equation B16, and Equation B15 can be gathered in matricial form to relate the different Jacobians between the source point and the undisplaced destination point:

$$\begin{bmatrix} \frac{\partial \boldsymbol{F}_D}{\partial \boldsymbol{u}_D} & \frac{\partial \boldsymbol{F}_D}{\partial \boldsymbol{\theta}_D} \\ \frac{\partial \boldsymbol{M}_D}{\partial \boldsymbol{u}_D} & \frac{\partial \boldsymbol{M}_D}{\partial \boldsymbol{\theta}_D} \end{bmatrix}_{\text{undisplaced}} = \begin{bmatrix} \boldsymbol{I} & \boldsymbol{0} \\ \tilde{\boldsymbol{s}}_0 & \boldsymbol{I} \end{bmatrix} \begin{bmatrix} \frac{\partial \boldsymbol{F}_S}{\partial \boldsymbol{u}_S} & \frac{\partial \boldsymbol{F}_S}{\partial \boldsymbol{\theta}_S} \\ \frac{\partial \boldsymbol{M}_S}{\partial \boldsymbol{u}_S} & \frac{\partial \boldsymbol{M}_S}{\partial \boldsymbol{\theta}_S} \end{bmatrix} \begin{bmatrix} \boldsymbol{I} & -\tilde{\boldsymbol{s}}_0 \\ \boldsymbol{0} & \boldsymbol{I} \end{bmatrix} + \begin{bmatrix} \boldsymbol{0} & \boldsymbol{0} \\ -\tilde{\boldsymbol{F}}_{S0} & \tilde{\boldsymbol{F}}_{S0}\tilde{\boldsymbol{s}}_0 \end{bmatrix} \tag{B17}$$

**Moments at the displaced destination point**

In this section, the moments are transferred to the displaced destination point. The vector from the displaced destination point to the displaced source is:

$$\boldsymbol{r} = \boldsymbol{s}_0 + \delta\boldsymbol{u}_S - \delta\boldsymbol{u}_D = \boldsymbol{s}_0 - \tilde{\boldsymbol{s}}_0\delta\boldsymbol{\theta}_S \tag{B18}$$

Introducing Equation B6 and Equation B18 into Equation B2, and temporarily using the "$\times$" notation instead of the tilde notation:

$$\boldsymbol{M}_{D0} + \delta\boldsymbol{M}_D = \boldsymbol{M}_{S0} + \delta\boldsymbol{M}_S + \boldsymbol{s}_0 \times \boldsymbol{F}_{S0} + \boldsymbol{s}_0 \times \delta\boldsymbol{F}_S - (\boldsymbol{s}_0 \times \delta\boldsymbol{\theta}_S) \times \boldsymbol{F}_{S0} - (\boldsymbol{s}_0 \times \delta\boldsymbol{\theta}_S) \times \delta\boldsymbol{F}_S \tag{B19}$$

Making use of Equation B8, neglecting the nonlinear term $((\boldsymbol{s}_0 \times \delta\boldsymbol{\theta}_S) \times \delta\boldsymbol{F}_S)$, and reintroducing the tilde notation leads to:

$$\delta\boldsymbol{M}_D = \delta\boldsymbol{M}_S + \tilde{\boldsymbol{s}}_0\delta\boldsymbol{F}_S + \tilde{\boldsymbol{F}}_{S0}\tilde{\boldsymbol{s}}_0\delta\boldsymbol{\theta}_S \tag{B20}$$

The Jacobians of the loads at the displaced node $D$ with respect to the displacements at node $D$ are then obtained by applying the chain rule to Equation B20 and making use of the Jacobian of the displacements given on the right of Equation B4.

$$\frac{\partial \boldsymbol{M}_D}{\partial \boldsymbol{u}_D} = \frac{\partial \boldsymbol{M}_S}{\partial \boldsymbol{u}_S}\frac{\partial \boldsymbol{u}_S}{\partial \boldsymbol{u}_D} + \frac{\partial \boldsymbol{M}_S}{\partial \boldsymbol{\theta}_S}\frac{\partial \boldsymbol{\theta}_S}{\partial \boldsymbol{u}_D} + \tilde{\boldsymbol{s}}_0\frac{\partial \boldsymbol{F}_S}{\partial \boldsymbol{u}_S}$$

$$= \frac{\partial \boldsymbol{M}_S}{\partial \boldsymbol{u}_S} + \tilde{\boldsymbol{s}}_0\frac{\partial \boldsymbol{F}_S}{\partial \boldsymbol{u}_S} \tag{B21}$$

and

$$\frac{\partial \boldsymbol{M}_D}{\partial \boldsymbol{\theta}_D} = \frac{\partial \boldsymbol{M}_S}{\partial \boldsymbol{u}_S}\frac{\partial \boldsymbol{u}_S}{\partial \boldsymbol{\theta}_D} + \frac{\partial \boldsymbol{M}_S}{\partial \boldsymbol{\theta}_S}\frac{\partial \boldsymbol{\theta}_S}{\partial \boldsymbol{\theta}_D} + \tilde{\boldsymbol{s}}_0\frac{\partial \boldsymbol{F}_S}{\partial \boldsymbol{\theta}_D} + \tilde{\boldsymbol{F}}_{S0}\tilde{\boldsymbol{s}}_0$$

$$= \frac{\partial \boldsymbol{M}_S}{\partial \boldsymbol{\theta}_S} - \frac{\partial \boldsymbol{M}_S}{\partial \boldsymbol{u}_S}\tilde{\boldsymbol{s}}_0 + \tilde{\boldsymbol{s}}_0\frac{\partial \boldsymbol{F}_S}{\partial \boldsymbol{\theta}_D} + \tilde{\boldsymbol{F}}_{S0}\tilde{\boldsymbol{s}}_0$$

$$= \frac{\partial \boldsymbol{M}_S}{\partial \boldsymbol{\theta}_S} - \frac{\partial \boldsymbol{M}_S}{\partial \boldsymbol{u}_S}\tilde{\boldsymbol{s}}_0 + \tilde{\boldsymbol{s}}_0\frac{\partial \boldsymbol{F}_S}{\partial \boldsymbol{\theta}_S} - \tilde{\boldsymbol{s}}_0\frac{\partial \boldsymbol{F}_S}{\partial \boldsymbol{u}_S}\tilde{\boldsymbol{s}}_0 + \tilde{\boldsymbol{F}}_{S0}\tilde{\boldsymbol{s}}_0 \tag{B22}$$

**Jacobian relationships at the displaced destination point**

Equation B11, Equation B22, and Equation B21 can be gathered in matricial form to relate the different Jacobians:

$$\begin{bmatrix} \frac{\partial \boldsymbol{F}_D}{\partial \boldsymbol{u}_D} & \frac{\partial \boldsymbol{F}_D}{\partial \boldsymbol{\theta}_D} \\ \frac{\partial \boldsymbol{M}_D}{\partial \boldsymbol{u}_D} & \frac{\partial \boldsymbol{M}_D}{\partial \boldsymbol{\theta}_D} \end{bmatrix}_{\text{displaced}} = \begin{bmatrix} \boldsymbol{I} & \boldsymbol{0} \\ \tilde{\boldsymbol{s}}_0 & \boldsymbol{I} \end{bmatrix} \begin{bmatrix} \frac{\partial \boldsymbol{F}_S}{\partial \boldsymbol{u}_S} & \frac{\partial \boldsymbol{F}_S}{\partial \boldsymbol{\theta}_S} \\ \frac{\partial \boldsymbol{M}_S}{\partial \boldsymbol{u}_S} & \frac{\partial \boldsymbol{M}_S}{\partial \boldsymbol{\theta}_S} \end{bmatrix} \begin{bmatrix} \boldsymbol{I} & -\tilde{\boldsymbol{s}}_0 \\ \boldsymbol{0} & \boldsymbol{I} \end{bmatrix} + \begin{bmatrix} \boldsymbol{0} & \boldsymbol{0} \\ \boldsymbol{0} & \tilde{\boldsymbol{F}}_{S0}\tilde{\boldsymbol{s}}_0 \end{bmatrix} \tag{B23}$$

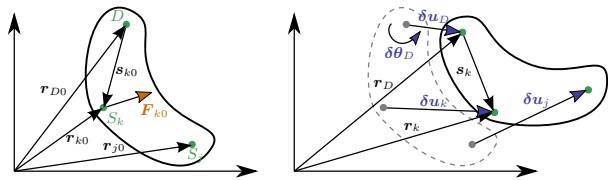

**Figure B2.** Rigid-body kinematics with the loads from multiple source points ($S_j$), transferred to a destination point ($D$)

## B2 Relationships at the displaced destination point for multiple source points
We now consider the case where multiple point sources are present. The derivation can be seen as a generalization of the previous case
between two points, but special care is needed when summing the contributions from the different nodes. The notations are illustrated in
Figure B2. The loads at the destination points are obtained as:
$$\boldsymbol{F}_D = \sum_k \boldsymbol{F}_k \tag{B24}$$

$$\boldsymbol{M}_D = \sum_k \boldsymbol{M}_k + \tilde{\boldsymbol{s}}_{\boldsymbol{k}} \boldsymbol{F}_k \tag{B25}$$

where $k$ is an index looping over all points of the rigid structure. To shorten notations, we define the vector between the destination point and
a given point as:
$$\boldsymbol{s}_k = \boldsymbol{r}_k - \boldsymbol{r}_D \tag{B26}$$

$$\boldsymbol{s}_{k0} = \boldsymbol{r}_{k0} - \boldsymbol{r}_{D0} \tag{B27}$$

where $\boldsymbol{s}_k$ is the vector between the displaced points and $\boldsymbol{s}_{k0}$ is the vector prior to the displacement. Due to the rigid-body assumption, the
elementary displacements of the points are related as follows:
$$\delta \boldsymbol{u}_D = \delta \boldsymbol{u}_j + \tilde{\boldsymbol{s}}_{\boldsymbol{j0}} \delta \boldsymbol{\theta}_j$$

$$\delta \boldsymbol{\theta}_D = \delta \boldsymbol{\theta}_j \tag{B28}$$

from which one obtains the following relationships:
$$\frac{\partial \boldsymbol{u}_j}{\partial \boldsymbol{u}_D} = \boldsymbol{I}, \quad \frac{\partial \boldsymbol{\theta}_j}{\partial \boldsymbol{u}_D} = \boldsymbol{O}, \quad \frac{\partial \boldsymbol{u}_j}{\partial \boldsymbol{\theta}_D} = -\tilde{\boldsymbol{s}}_{\boldsymbol{j0}}, \quad \frac{\partial \boldsymbol{\theta}_j}{\partial \boldsymbol{\theta}_D} = \boldsymbol{I}, \quad \frac{\partial \boldsymbol{\theta}_j}{\partial \boldsymbol{\theta}_k} = \boldsymbol{I}\delta_{jk}, \quad \frac{\partial \boldsymbol{\theta}_j}{\partial \boldsymbol{u}_k} = \boldsymbol{O} \tag{B29}$$

Using a similar Taylor expansion as for the case with two nodes, the perturbation loads are obtained as:
$$\delta \boldsymbol{F}_D = \sum_k \delta \boldsymbol{F}_k \tag{B30}$$

$$\delta \boldsymbol{M}_D = \sum_k \delta \boldsymbol{M}_k + \tilde{\boldsymbol{s}}_{\boldsymbol{k0}} \delta \boldsymbol{F}_k + \tilde{\boldsymbol{F}}_{k0}(\tilde{\boldsymbol{s}}_{\boldsymbol{k0}} \delta \boldsymbol{\theta}_k) \tag{B31}$$

The chain rule for a given quantity of interest ($Q$) is obtained by summing over all the elementary variables:
$$dQ = \sum_j \frac{\partial Q}{\partial \boldsymbol{u}_j} d\boldsymbol{u}_j + \frac{\partial Q}{\partial \boldsymbol{\theta}_j} d\boldsymbol{\theta}_j \tag{B32}$$

For instance, applying the chain rule to $\boldsymbol{F}_D$ and using Equation B30 leads to:
$$\frac{\partial \boldsymbol{F}_D}{\partial \boldsymbol{u}_D} = \sum_j \frac{\partial \boldsymbol{F}_D}{\partial \boldsymbol{u}_j} \frac{\partial \boldsymbol{u}_j}{\partial \boldsymbol{u}_D} + \frac{\partial \boldsymbol{F}_D}{\partial \boldsymbol{\theta}_j} \frac{\partial \boldsymbol{\theta}_j}{\partial \boldsymbol{u}_D} = \sum_j \sum_k \frac{\partial \boldsymbol{F}_k}{\partial \boldsymbol{u}_j} \frac{\partial \boldsymbol{u}_j}{\partial \boldsymbol{u}_D} + \frac{\partial \boldsymbol{F}_k}{\partial \boldsymbol{\theta}_j} \frac{\partial \boldsymbol{\theta}_j}{\partial \boldsymbol{u}_D} = \sum_j \sum_k \frac{\partial \boldsymbol{F}_k}{\partial \boldsymbol{u}_j} \tag{B33}$$

Eventually, the Jacobians at the displaced destination node are obtained as:
$$\begin{bmatrix} \frac{\partial \boldsymbol{F}_D}{\partial \boldsymbol{u}_D} & \frac{\partial \boldsymbol{F}_D}{\partial \boldsymbol{\theta}_D} \\ \frac{\partial \boldsymbol{M}_D}{\partial \boldsymbol{u}_D} & \frac{\partial \boldsymbol{M}_D}{\partial \boldsymbol{\theta}_D} \end{bmatrix}_{\text{displaced}} = \sum_j \left\{ \sum_k \left( \begin{bmatrix} \boldsymbol{I} & \boldsymbol{0} \\ \tilde{\boldsymbol{s}}_{k0} & \boldsymbol{I} \end{bmatrix} \begin{bmatrix} \frac{\partial \boldsymbol{F}_k}{\partial \boldsymbol{u}_j} & \frac{\partial \boldsymbol{F}_k}{\partial \boldsymbol{\theta}_j} \\ \frac{\partial \boldsymbol{M}_k}{\partial \boldsymbol{u}_j} & \frac{\partial \boldsymbol{M}_k}{\partial \boldsymbol{\theta}_j} \end{bmatrix} \begin{bmatrix} \boldsymbol{I} & -\tilde{\boldsymbol{s}}_{j0} \\ \boldsymbol{0} & \boldsymbol{I} \end{bmatrix} \right) + \begin{bmatrix} \boldsymbol{0} & \boldsymbol{0} \\ \boldsymbol{0} & \tilde{\boldsymbol{F}}_{j0}\tilde{\boldsymbol{s}}_{j0} \end{bmatrix} \right\} \tag{B34}$$

## Appendix C: Verification of the linear models

In this section, we supplement the results given in subsubsection 3.3.3 by showing free-decay results without hydrodynamics (no added
mass, damping, hydrostatics). In Figure C1, we show results with the structure only, and results with the structure and moorings are reported

in Figure C2. These results also include the nonlinear WELIB formulation. A strong agreement is found between the nonlinear OpenFAST

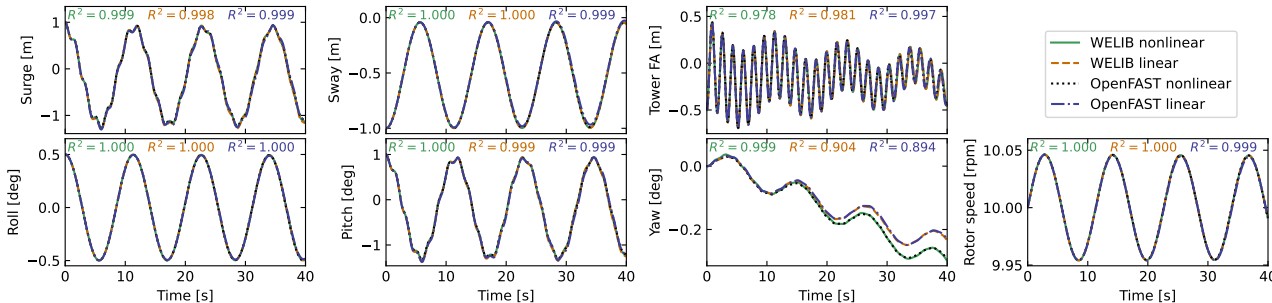

**Figure C1.** Free decay of the structure using nonlinear and linear models for a case including only the structure (no moorings, no hydrodynamics). Time series of the main DOF.

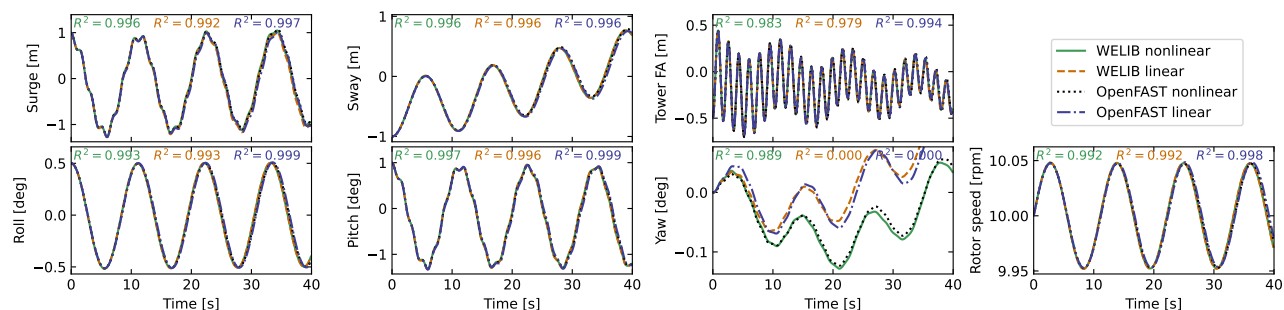

**Figure C2.** Free decay of the structure using nonlinear and linear models for a case including moorings (no hydrodynamics). Time series of the main DOF.


and WELIB models, and between the linear OpenFAST and WELIB models. The yaw degree of freedom appears to be more challenging
to capture for the linear models. We recall that OpenFAST and WELIB use different definition of the transformation matrices between the
degrees of freedom, which results in different structural dynamic equations.

## Appendix D: Computation of section loads

In this section, we describe the nonlinear calculation procedure used in subsubsection 3.6.2 to assess the section loads along the tower based on estimates of the structure kinematics and the loads at the tower top. For conciseness, in this appendix, we use $x$ and $z$ for the coordinates along the tower fore-aft and tower height, respectively, instead of $x_T$ and $z_T$.

### D1 Tower fore-aft bending moment and shear force

The fore-aft and side-side moments are computed in the same way; therefore, this section focuses on the fore-aft direction. The sectional fore-aft bending moment at a given tower height $z$ is determined as:

$$\mathcal{M}_y(z) = \mathcal{M}_{y,\text{top}} - \int_z^{L_T} S_x(z')dz' \tag{D1}$$

Here, $\mathcal{M}_{y,\text{top}}$ is the fore-aft bending moment at the tower top, and $S_x$ is the shear force in the $x$ direction, obtained as:

$$S_x(z) = \int_z^{L_T} p_{x,\text{all}}(z')\,dz' \tag{D2}$$

where $p_{x,\text{all}}$ is the force per unit length acting on the tower section in the fore-aft direction, including contributions from the external loads (aerodynamic loads on the structure), inertial loads due to the acceleration of the structure (including gravity), and nonlinear correction terms from the loads in the $z$ direction ($p - \Delta$ effect, including self-weight effects). The different contributions are written as follows:

$$p_{x,\text{all}} = p_{x,\text{ext}} + p_{x,\text{corr}} - p_{x,\text{acc}} \tag{D3}$$

In this work, we neglect the external loads on the tower, $p_{x,\text{ext}} = 0$ (aerodynamic loads on the tower are typically small relative to rotor-thrust loads for an operating wind turbine). The acceleration contribution is $p_{x,\text{acc}} = -m(z)(a_{x,\text{struct}}(z) - a_{x,\text{grav}})$, where $m$ is the mass per length along the beam, $a_{x,\text{struct}}(z)$ is the acceleration of the section, determined based on the rigid-body acceleration of the floater and the elastic motion of the tower ($\dot{q}_T$ and $\ddot{q}_T$), and $a_{x,\text{grav}}$ is the acceleration of gravity in the $x$ direction. The $p - \Delta$ correction term due to the vertical loading is computed as (see Branlard (2019)):

$$p_{x,\text{corr}} = \frac{d^2\Phi}{d^2z}\left[\int_z^{L} p_z\,dx' + \sum_{z_k \geq z} \mathcal{F}_{z,k}\right] - \frac{d\Phi}{dz}\left[p_z + \sum_k \mathcal{F}_{z,k}\delta(z - z_k)\right] \tag{D4}$$

where $p_z$ is the vertical load per unit length (mostly consisting of the self-weight), $\mathcal{F}_{z,k}$ is the $k$th vertical force acting at point $z_k$, $\delta$ is the Dirac function, and $\Phi$ is the shape function used to describe the tower displacement field (see subsubsection 2.3.3). In our case, only the vertical force acting on top of the tower is present, $z_1 = L_T$ and $\mathcal{F}_{z,1} = \mathcal{F}_{z,\text{top}}$. The procedure to compute the section loads in the $y$ direction (using the $p - \Delta$ correction as well) is similar.

### D2 Tower and rotor-nacelle assembly kinematics

The determination of the tower section loads requires a knowledge of the tower kinematics, to compute $\boldsymbol{a}_{\text{struct}}$, and of the rotor-nacelle assembly (RNA) kinematics, to compute the inertial contribution to the tower-top loads (see subsection D3). The position, linear velocity,

linear acceleration, rotational speed, and rotational acceleration of the floater (point $F$, body $f$) are given respectively by:

$$\boldsymbol{r}_F = \{x, y, z\}_i, \quad \boldsymbol{v}_F = \{\dot{x}, \dot{y}, \dot{z}\}_i, \quad \boldsymbol{a}_F = \{\ddot{x}, \ddot{y}, \ddot{z}\}_i, \tag{D5}$$

$$\boldsymbol{\omega}_f = \{\dot{\phi}_z, \dot{\phi}_y, \dot{\phi}_z\}_i, \quad \dot{\boldsymbol{\omega}}_f = \{\ddot{\phi}_z, \ddot{\phi}_y, \ddot{\phi}_z\}_i \tag{D6}$$

where the notation $i$ indicates that the coordinates of the vector are expressed in the inertial coordinate system. The transformation matrix
from the floater to the inertial frame is obtained as $\boldsymbol{R}_{f2i} = \boldsymbol{R}(\phi_x, \phi_y, \phi_z)$, where $\boldsymbol{R}$ is a function computing the rotation matrix. The tower
base (point $T$, body $t$) kinematics are obtained from the floater motion using rigid-body kinematics:

$$\boldsymbol{r}_T = \boldsymbol{r}_F + \boldsymbol{r}_{FT}, \tag{D7}$$

$$\boldsymbol{v}_T = \boldsymbol{v}_F + \boldsymbol{\omega}_f \times \boldsymbol{r}_{FT}, \tag{D8}$$

$$\boldsymbol{a}_T = \boldsymbol{a}_F + \boldsymbol{\omega}_f \times (\boldsymbol{\omega}_f \times \boldsymbol{r}_{FT}) + \dot{\boldsymbol{\omega}}_f \times \boldsymbol{r}_{FT}, \tag{D9}$$

$$\boldsymbol{\omega}_t = \boldsymbol{\omega}_f, \quad \dot{\boldsymbol{\omega}}_t = \dot{\boldsymbol{\omega}}_f, \quad \boldsymbol{R}_{t2i} = \boldsymbol{R}_{f2i} \tag{D10}$$

742  where $\boldsymbol{r}_{FT}$ is the vector from the floater point to the tower base. The kinematics of a given tower section (point $S$, at height $z$) are given by:

$$\boldsymbol{r}_S = \boldsymbol{r}_T + \boldsymbol{r}_{TS} = \boldsymbol{r}_T + \boldsymbol{r}_{TS_0} + \boldsymbol{u}_S, \tag{D11}$$

$$\boldsymbol{v}_S = \boldsymbol{v}_T + \boldsymbol{\omega}_t \times \boldsymbol{r}_{TS} + \dot{\boldsymbol{u}}_S, \tag{D12}$$

$$\boldsymbol{a}_S = \boldsymbol{a}_T + \boldsymbol{\omega}_t \times (\boldsymbol{\omega}_t \times \boldsymbol{r}_{TS}) + \dot{\boldsymbol{\omega}}_t \times \boldsymbol{r}_{TS} + 2\boldsymbol{\omega}_t \times \dot{\boldsymbol{u}}_S + \ddot{\boldsymbol{u}}_S, \tag{D13}$$

$$\boldsymbol{\omega}_s = \boldsymbol{\omega}_t + \boldsymbol{\omega}_{ts}, \tag{D14}$$

$$\dot{\boldsymbol{\omega}}_s = \dot{\boldsymbol{\omega}}_t + \dot{\boldsymbol{\omega}}_{ts} + \boldsymbol{\omega}_t \times \boldsymbol{\omega}_{ts}, \tag{D15}$$

where $\boldsymbol{r}_{TS_0} = z\hat{\boldsymbol{z}}_t$ is the vector from the tower base to the undeflected section, $\boldsymbol{u}_S, \dot{\boldsymbol{u}}_S, \ddot{\boldsymbol{u}}_S$ are the elastic motions of the section computed
based on the shape function and the generalized coordinates, e.g., $\boldsymbol{u}_S(z) = \sum_j q_{t,j} \boldsymbol{\Phi}_j = q_t \Phi(z)\hat{\boldsymbol{x}}_t$ (see Branlard and Geisler (2022)). We
note that OpenFAST also includes a vertical motion associated with the deflection (referred to as a "geometric nonlinearity"), which we
neglect in this work. The transformation matrix from the section to the tower is $\boldsymbol{R}_{s2t} = \boldsymbol{R}(-u'_{S,y}, u'_{S,x}, 0)$, where $u_{S,y}$ and $u_{S,x}$ are the
components of $\boldsymbol{u}_S$ in the tower coordinate system, and the prime notation indicates the differentiation with respect to $z$. The rotation speed
and acceleration of the tower section with respect to the tower base are:

$$\boldsymbol{\omega}_{ts} = \{\dot{u}'_{S,y}, \dot{u}'_{S,x}, 0\}_t, \quad \dot{\boldsymbol{\omega}}_{ts} = \{\ddot{u}'_{S,y}, \ddot{u}'_{S,x}, 0\}_t \tag{D16}$$

The kinematics of the tower-top point and nacelle (point $N$, body $n$) are taken from the last section node (point $S$ with $z = L_T$). Yawing,
tilting, and rolling of the tower top would change the orientation matrix, rotational velocity, and rotational acceleration of the nacelle. These
kinematics are omitted here for conciseness. The kinematics of the center of mass of the RNA (point $G$) are obtained using rigid-body
kinematics, identical to what was used between point $F$ and $T$.

## D3   Tower-top loads

The tower-top loads are computed as follows:

$$\boldsymbol{\mathcal{F}}_{\text{top}} = \boldsymbol{\mathcal{F}}_{\text{aero}} - \boldsymbol{\mathcal{F}}_{\text{inertia}} \tag{D17}$$

$$\boldsymbol{\mathcal{M}}_{\text{top}} = \boldsymbol{\mathcal{M}}_{\text{aero}} - \boldsymbol{\mathcal{M}}_{\text{inertia}} \tag{D18}$$

where the aerodynamic loads are transferred to the tower top and where the inertial loads from the rigid-body RNA are:

$$\mathcal{F}_{\text{inertia}} = M_{\text{RNA}}(\boldsymbol{a}_G - \boldsymbol{g}) \tag{D19}$$

$$\mathcal{M}_{\text{inertia}} = \boldsymbol{r}_{NG} \times \mathcal{F}_{\text{inertia}} + \boldsymbol{J}_G \cdot \dot{\boldsymbol{\omega}}_n + \boldsymbol{\omega}_n \times (\boldsymbol{J}_G \cdot \boldsymbol{\omega}_n) \tag{D20}$$

where: $\boldsymbol{r}_{NG}$ is the vector from the tower top to the center of mass of the RNA, $M_{\text{RNA}}$ is the mass of the RNA, $\boldsymbol{J}_G$ is the inertia tensor of the RNA at it's center of mass, $\boldsymbol{a}_G$ is the linear acceleration of the center of mass of the RNA, $\boldsymbol{\omega}_n$ is the rotational acceleration of the RNA, and $\dot{\boldsymbol{\omega}}_n$ is the rotational acceleration of the nacelle. The load calculation is first done in the coordinate system of the nacelle and then transferred to the coordinate system of the tower where Equation D1 is defined.

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
