# Peer review of "A digital twin solution for floating offshore wind turbines validated using a full-scale prototype"

_Wind Energy Science, 2023_

## Referee Comment (RC1)

469-2017, 2017.

[referee-annotated manuscript omitted]

---

## Author Comment (AC1)

Dear reviewers

Thank you so much for your time and effort in reviewing our paper. Please find our answers to your comments and the corresponding updates to the manuscript below. We hope to have addressed most of your comments which helped us improve the manuscript, and we will be happy to continue the discussion. At the end of this PDF, you can find the differences made to the manuscript with highlighted colors. In the coming days, we will submit the revised manuscript.

Emmanuel and co-authors

**Reviewer 1**

The paper addresses the "construction" of a digital twin for floating wind turbines that is verified against numerical simulations in a state-of-the-art offshore code and validated against real-world data. At present, the first commercial floating wind turbines are being installed and digital tools as the one presented in this article can be fundamental for their success. Therefore, the article is relevant to the international scientific community and falls within the scope of WES.

In general, the objectives and hypotheses of the article are clear. The methods are valid and can be reproduced by the reader, also thanks to the fact that the source code of the digital twin will be shared in a GitHub repository once the article will be published. The analyses presented by authors are valid and backed up by widely used simulations tools and previous research. Results are outlined in a clear way and the article is generally well structured in all its parts (text, figures, mathematics).

For all this reasons, the article deserves to be considered for publication in WES after being revised.

As a general comment, I see the tools presented in this article are derived from those developed in previous works about digital twins for bottom-fixed turbines. I suggest the authors to point out in the introduction the additional challenges of making a digital tool for a wind turbine with floating foundations. It may sound trivial, but I would explain why one cannot simply use tools derived for bottom-fixed turbines and where these are inaccurate.

>> Thank you for your comment, we have added the following paragraph in the introduction to highlight some of the challenges of the floating case:

"Developing digital twins for floating wind turbines present a set of challenges compared to our previous work on fixed-bottom foundations. The potentially large motions undergone by the platform may effect the aerodynamics and accelerometer signals. The models developed for fixed-bottom foundations need to be augmented to be able to predict the aerodynamics when the platform experiences large pitching motions. The dynamics of the platform motion needs to be well captured for the tower-top accelerometer to be used and for estimating the loading in the stationkeeping system. In both floating and fixed-bottom wind turbines, hydrodynamic loads need to be estimated to capture member-level loads in the substructure but they can be omitted as a first approximation if only the tower loads are estimated, as in this study."

Specific comments are reported below, and technical corrections are in the attached document.

Specific comments

102-103. The OpenFAST model is complemented with a closed-loop controller: is it based on a reference controller like ROSCO, are you allowed to give the reader some information about it?

The controller parameters are tuned so that OpenFAST simulations match SCADA data for selected operating conditions. Can you add something about the controller tuning?

>>> Thank you for your comment, we have modified the text as follows to give more details on the controller and its tuning:

"The OpenFAST model is complemented with NREL's Reference OpenSource Controller (ROSCO, Abbas et al. (2021)). The controller parameters are tuned so that OpenFAST simulations match the operating conditions of the turbine extracted from SCADA data (pitch, rotor speed and power). The nacelle velocity feedback option of ROSCO is used to reduce the platform pitching motion. Using trial and error, the frequency and damping ratio of the pitch PI-controller are set to $\omega p = 0.05$ rad/s and $\zeta p = 7\%$, and the values for the torque controller are set $\omega Q = 0.15$ rad/s and $\zeta Q = 7\%$. The gain-scheduling of the pitch controller are obtained using the tuning feature of ROSCO. We note that the controller is only needed to perform verifications of the digital twin with realistic time series of the turbine responses, but the controller itself is not used for the design of the digital twin."

l 109: why is it challenging?

>>> We have added the following to the text: "because the variations of the wind speed are sudden"

l 123

Can you say something more about the assumption of rotational symmetry of the platform? How do you think it might affect your study and its results?

>>> This is indeed a crude assumption. We have performed a sensitivity analysis on the mass and stiffness matrix and found that the assumption remained fair. We have added the following paragraph in the article

"Some of the consequences of this assumption is that we do no capture changes of inertial properties due to asymmetry of the support structure and changes of stiffness of the mooring system. In the case of the TetraSpar, the mass matrix of the floater does not vary significantly with the yawing of the coordinate system, and the assumption appears fair. We note that if the structure had perfect 120 \unit{deg} symmetry about the yaw axis, then its inertia would be invariant by yaw rotation. For the restoring stiffness of the mooring system, the diagonal terms do not vary significantly as the coordinate system yaws, but some of the coupling terms vary by 50\% to 200\%. The couplings between the platform degrees of freedom are likely wrongly estimated under the rotational symmetry assumption. The impact is nevertheless limited because most of the platform DOF ($x$, $y$, $\phi_x$, and $\phi_y$) are measured and therefore observable by the Kalman filter."

137-138. The capabilities of the Python code are not clear. If important for the rest of the study, I would explain better how the Python tools work and how it carries out the operations listed at lines 137-138

>>> Thank you for your comment, we have tried to reformulate the paragraph to highlight the features of the python code, and which ones are used.

"The WELIB approach consists of a set of dedicated open-source Python tools that are similar to the ElastoDyn, HydroDyn and MAP modules of OpenFAST. We developed these tools to offer additional modularity and granularity: the tools can be called individually or together; their states, inputs and outputs can be accessed and manipulated at each time step; and the structural dynamics equations are obtained analytically. For instance, this allows for: 1) analytical linearization of the structural dynamics, 2) simple linearization of the hydrodynamics (obtention of $6 \times 6$ matrices), 3) linearization of hydrodynamics with respect to wave elevation, 4) linearization with respect to parameters (Jonkman et al., 2022), and 5) interactive time-stepping of the linear and nonlinear model. In this work, we mostly use the first two features listed above and their usage will be described in Section 3.3.2. Results from time-stepping simulations will be presented in Section 3.3.3."

139-147. This paragraph does not convey adequately the difference between OpenFAST and WELIB. I understand that: 1) WELIB is somehow simpler (and less accurate?) than OpenFAST for nonlinear simulations 2) WELIB is easier to linearize and since it is based on an analytical approach it provides better understanding of the wind turbine physics.

I suggest the authors to work a bit on this paragraph to help the reader understand the complementarity of the two simulation tools with respect to the objectives of this study.

>>> Based on your multiple comments on this paragraph, we have tried to rewrite it. To make it clearer, we have split it into three parts: an overview, a description of WELIB tools, and a paragraph on the differences between the two.

Here are some excerpts of the rewritten paragraph:

Overview:

" In the next sections, we will show that the results from both approaches are consistent with each other so that either of the two can be used to obtain nonlinear and linear reduced order models. Ultimately, in Section 4, a mix of the two approaches is used for the digital twin: linear OpenFAST models for the state-space equations (Section 3.5) and WELIB for the virtual sensing step (Section 3.6)."

Comparison:

"Currently, no controller or aerodynamic module is present in WELIB. Therefore, nonlinear timestepping simulations with WELIB are limited to free-decay simulations or prescribed loads. Another shortcoming is that WELIB does not cover the full range of options available with OpenFAST, which is a continuously evolving, extensively verified and validated tool. Such options include the potential flow representation of hydrodynamic bodies, the flexibility of the floating structure, aerodynamic and control features. One benefit of WELIB over OpenFAST is the possibility to perform interactive time-stepping, that is, to change the states and inputs dynamically during the simulation. We do not use this approach in this work, but it can be considered for nonlinear digital twin applications, for instance, using an extended Kalman filter algorithm. Another benefit is the possibility to obtain analytical linear models of the structure, which avoids using finite-differences and therefore reduces the associated numerical errors. In the WELIB approach, the individual modules are linearized separately before being combined into the final linear model, and it is therefore easier to understand where each term in the Jacobians of the linear models comes from, and thereby, gain physical intuitiveness on the model. Ultimately, the linear models obtained by both approaches are similar and differ mostly based on differences in the structural dynamics equations and the implementation of rotational transformation matrices."

146, "to obtain analytical linear models and gain physical intuitiveness on the model". From this sentence it's not clear the difference between a linear model obtained in WELIB and in OpenFAST. In the previous sentence it is said that WELIB can perform nonlinear simulations, so the difference for the linearized model is in how the model is obtained. I think you should briefly mention this issue and introduce what is explained in the next sections.

>>> Thank you for pointing this out, the models indeed mostly differ in how they are obtained. We hope that the rewritten paragraph clarifies this better.

169, "in two different ways". It's not clear the added value of performing the linearization with Python and the HydroDyn module. I suggest the authors to focus on one of the two, or clarify why it is important to show in this study the results obtained with the two methodologies.

>>> Thank you for pointing that out. We have now rewritten this paragraph and tried to highlight the advantages and inconvenient of the different approaches. Obtaining the linearized hydrodynamic matrices is a question that often is asked by the OpenFAST community, and we therefore experimented with different approaches. The first one we tried (using full-system linearization) ended up being quite an involved process and somehow an important part of this work. We have therefore chosen to include it in this paper, as we do not think the approach is published elsewhere. The approach ended up being quite lengthy and prone to errors, which is why we turned to the other approach, simpler to implement and faster to run. We have tried to make the description less confusing in the updated version. The updated paragraph is as follows:

"We compute the 6×6 linearized rigid-body hydrodynamics matrices (mass matrix Mh, damping matrix Ch, and stiffness matrix Kh) corresponding to the six rigid-body motions of the platform. At the time of this study, these matrices could not be obtained directly from OpenFAST. While working on this issue, we ended up devising multiple ways to obtain them. They can now be obtained using: 1) full-system linearization of the HydroDyn module, 2) the Python implementation of the HydroDyn module by performing rigid-body perturbations of the full platform, or 3) an upgraded version of the OpenFAST HydroDyn driver that also uses rigid-body perturbations. The first approach uses baseline OpenFAST functionalities but requires additional postprocessing scripts and derivations. The full-system linearization of OpenFAST provides Jacobians of the hydrodynamic loads as a function of motions of the individual hydrodynamic analysis nodes (of which models often have hundreds to thousands of). To transfer these individual Jacobians to the reference point and obtain the 6×6 matrices, we developed and used the method presented in Appendix B. The process is involved and prone to errors. In comparison, the second and third approaches are straightforward to implement and are several orders of magnitude faster. The Python version was implemented first and then ported over to Fortran so that it can be readily available to the OpenFAST community. The consistency between the different approaches was verified, and because of its ease of use, the second approach is retained in this study. We note that in this study, all members are modeled using the Morison equation and the hydrodynamic drag is set to zero during the linearization process. There is therefore no frequency-dependent damping, and the effect of hydrodynamic drag is assumed to be part of the modeling uncertainty of the state estimator (see Section 3.5)."

180-181. Not clear what is Q0. I think you should rember that the degress of freedom of Mh are the platform motions which are only a subset of q. You should also recall the dimensions of Q0 and its structure.

>>> Additional details are now provided in the sentence: "The matrix Q0, of dimension 8 × 6, maps the subset of the 6 rigid-body platform degrees of freedom (x,y,z,φx,φy,φz), used for the definitions of Mh, Ch, Kh and Km, to the full vector of degrees of freedom, q."

3.3 Verification of the linear models. I suggest introducing this paragraph explaining why you used free decay simulations for verification of the linear models and no other types of simulation. Moreover, which is the purpose of simulations without hydrostatic restoring?

>>> Thank you for your comment, these are indeed valid points that we were explaining. We have added the following in the paragraph to justify the use of free-decay simulation:

"Free-decay simulations are sufficient because wave and aerodynamic loads are purposely not included in the linear models used by the digital twin. "

Further, to justify the use of simulations with hydrostatics, we have added the following:

"First, simulation without hydrodynamics (structure only) are considered, to isolate and verify the structural-dynamics part of the models."

l357  How did you get this information?

>>> This was indeed not clear. We performed sensitivity analyses (perturbing the mean platform pitch, tower fore-aft and thrust by +/- 5%) and looked at the relative changes in the section loads. We modified the text as follows:

"After performing a sensitivity analysis on the inputs and states of the system, we observed that the variables that most affect the fore-aft section loads are the platform pitch ( $\phi_y$), the tower fore-aft bending degree of freedom ($q_t$), and the aerodynamic thrust."

383, "adequate filtering of the input measurements can be used to tune the energy content at high frequencies". Which is the impact of the frequency content above 1 Hz on fatigue loads? How meaningful it is compared to low-frequency harmonics? If you think they could be interesting, I suggest you add results of DEL obtained based on low-pass filtered signals.

>>> We believe there is quite some work to be done to further understand the differences of results between the DEL from the estimated signal and the reference signal. It seems that there is no easy way to get the DELs correctly because both low-frequencies and high-frequencies contribute to the DELs. For this paper, we prefer to point to our previous study, where indeed we showed that a tuned low-pass filter on the acceleration could lead to improvements in results. We have ideas on how this could be improved, but we would like to keep this for future work. We have added the following precision in the text: "As shown in previous studies, filtering of the input measurements can be used to tune the energy content at high frequencies. The method is yet unsatisfactory because it acts as an artificial rebalancing of energy content to achieve the desired DEL value. Both low and high frequency content contribute to the DEL values, therefore, we believe that systematic improvement is only possible through modeling improvements and higher observability of the states by the Kalman filter."

l 407: Can you give some numbers (min, max, mean)?

>>> We added the following precision in the text: "ranging from 4 to 24.5 \unit{m/s} with an overall mean of \unit{8.9} m/s."

421-422, "this is a crude first-order approximation". Can you add some words about what you are neglecting using the tower top load cell?

>>> Thank you for your comment. We have added the following precision in parenthesis to mention some of the differences between the aerodynamic thrust and the load measured at the tower top: "e.g., neglecting inertial and gravitational loads, nacelle tilting and shaft bending."

428, "… or that the state estimator is failing". Do you think that sensor issues can explain the different offset?

>>> You are correct, it could indeed be a source of error. We haven't found a clear explanation for the source of error. We have added the following in the text:

"Measurement errors could also affect the results, but no systematic error was detected over the time period investigated. It is therefore difficult to conclude as to what is the main source of error."

450, "gain-scheduling of the linear model, using nonlinear models". You should briefly explain how these techniques can improve the results you already got.

>>> Thank you for your comment. We have now added small explanations in the list of potential improvements in our conclusion.

"Additional improvements could include:

1) gain-scheduling of the linear models to extend the domain of validity of the linear models used and reduce the modeling error,

2) using nonlinear models and extended Kalman filtering techniques to lift the linear assumptions that challenges the aerodynamics, hydrodynamics and structural dynamics,

3) etc. "

**Reviewer 2**

Digital twins for floating wind turbines is a topic of interest lately within the wind energy community. This paper addresses this subject in a clear and elegant way that is easy to follow. It starts by introducing the building blocks separately before combining everything together to obtain the digital twin tool. Afterwards, measurements from numerical simulations and full-scale data of the TetraSpar prototype are used for the validation of the tool. Not only that the tool is validated, but also, every component forming the digital twin is verified before validating the whole thing.

The paper reveals the potential gains the digital twins have to offer, which further promotes the wind energy scientific community. The methods and analyses performed are clear and valid, the scientific approach is clearly explained, which helps with the repeatability of the work. There is a good balance between the methods and the results with proper referencing.

The results are well presented and properly discussed till arriving at a well-drawn conclusion summarizing the work done. The abstract efficiently conveys the main message of the paper.

Accordingly, the paper meets the criteria of the journal and qualifies for publication after revision.

I see my fellow referee have many similar comments to mine. So, for brevity, I will add a few more:

Can you explain why you used a model of 8 DOFs in particular? Why not more?

>>> Thank you for your comment, we have now added a small paragraph to justify the choice of degrees of freedom:

"The selected set of DOF capture the first-order effects as it is the minimal set required to capture the full motion of the floater (necessary to compute restoring loads and tower loads), the tower flexibility (necessary to capture tower loads) and the rotor motion (necessary to capture the aerodynamics). Additional degrees of freedom could be considered to increase the modeling accuracy, e.g., to include floater flexibility for internal calculation of floater loads. This would increase the computational requirement and only contribute to second-order effects, and it is therefore postponed to future work. "

You mentioned that controller tuning was performed to match the SCADA data for OpenFAST simulations. What about the control system in WELIB? Does it use a wrapper like pHydrodyn but for ServoDyn? Can you elaborate on that a bit more?

>>> Thank you for pointing that out. The statement that nonlinear simulation with WELIB can be compared with OpenFAST could be misleading because no controller is currently implemented in WELIB. We have therefore added the following in the text:

 "Currently, no controller or aerodynamic module is present in WELIB"

Lines 196-197, you mentioned that there exists a drift in the sway and roll responses, but you did not explain the reason for its existence. Can it be due to neglecting the hydrodynamic viscous term? Can it be the transformation matrix?

>>> Thank you for pointing this out. We do not have a "good explanation" for the drift. Free-decay simulations with less challenging initial conditions (smaller amplitudes) do no show such a drift, we therefore believe that the drift is mostly due to differences between linear and nonlinear models, which build up as you go away from the operating point. We have added the following at the end of the sentence: "which we assume can be attributed to inherent differences between linear and nonlinear models."

In Figure 4, there is an agreement in the peaks of the linear and nonlinear OpenFAST, while WELIB does not. Can you say something about that?

>>> We have revised this paragraph to further explain that the OpenFAST models are expected to be fully consistent (as they hit the same code base), whereas the difference between WELIB and OpenFAST linear are likely a propagation of the error in the yaw signal that couples to the other degrees of freedom.

"In all cases, the OpenFAST linear model is closer to the nonlinear OpenFAST model than the WELIB model. The consistency between the linear and nonlinear OpenFAST model is expected because they are obtained from the same code base. The WELIB linear model had difficulty capturing the yaw response. We believe that some of the error in the yaw signal is due to differences between the formulations of the three-dimensional rotations in OpenFAST and WELIB. The difference in yaw, results in a difference of coupling between the DOF, which can explain the differences observed in the sway, roll and rotor speed signals."

Line 389, can you cite a reference for the equations?

>>> There is no reference for these equations. A simple linear relationship was assumed based on the measurements at the given test site. We have modified the text to make this clearer:

"The $H\_s$ and $T\_p$ relationships were obtained by performing a linear regression on the sea state and wind measurements at the test site."

Line 451-452, "adding a model to account for wave excitation forces", can you elaborate a bit more what do you mean by this sentence? what would be the difference between this model and HydroDyn/ pHydrodyn?

>>> We have added the following paragraph higher up in the text when we discuss the hydrodynamic loads to clarify what we mean by "hydrodynamic model of the wave excitation force".

"In this work, δfh is mapped into the inherent model noise of the Kalman filter (see Section 3.5), but further improvements could be obtained by including a model for the wave-excitation loads. For instance, we could introduce a hydrodynamic state analog to the wave elevation or a set of states that scales different hydrodynamic shape functions so that the hydrodynamic load can be obtained as a linear superposition of scaled shape functions. In our application (tower section loads), such modeling did not appear necessary, but it will be considered in future work as it can be relevant to estimate substructure loads."

A final careful read should be done to brush off any spelling or grammatical mistakes.

>>> Thank you for your comment, we believe most of the spelling mistakes are now corrected.

**Reviewer 3**

This research focuses on the implementation of a physics-based digital twin solution for a floating offshore wind turbine, specifically the TetraSpar prototype. The digital twin aims to estimate aerodynamic loads, wind speed, and section loads along the tower to assess the fatigue lifetime of the structure. The solution incorporates a Kalman filter for structural state estimation, an aerodynamic estimator, and a physics-based virtual sensing procedure to obtain loads along the tower. The digital twin utilizes measurements commonly available on wind turbines and motion sensors typically used for floating platforms.

Two different approaches are explored to develop physics-based models: dedicated Python tools and the OpenFAST linearization feature, with a combination of both approaches in the final digital twin version. The OpenFAST linearization is for estimation of system states, and WELIB is used for sectional load calculations with the calculated states. Numerical experiments are conducted to verify the individual models, achieving estimated damage equivalent loads with an accuracy of 5% to 10% in simulation. When compared to measurements from the TetraSpar prototype, the errors increased to an average of 10%-15%. The results demonstrate promising accuracy and highlight the potential of digital twin solutions in estimating fatigue loads on floating offshore wind turbines.

Given the rise in popularity of digital twins for estimation of long term damage, I believe this article is fit for publication with minor adjustments and clarifications and will serve its purpose in continuing the progress of efficient digital twins for floating wind turbines.

For some general comments, I think the importance of computationally efficient digital twins should be highlighted and the computation time for each model should be evaluated, or explained.

Does OpenFAST compute fast enough for real time estimation?

If it doesn't, is it possible to assume a significant increase in computational power would allow it to be run in real time?

>>> Thank you for your comment. You are right that the computational time is indeed crucial to achieve the goal of online estimations. To mention this earlier in the text, we have added the following to the introduction:

"Achieving computational efficiency is crucial to be able to run the digital twin online, therefore, a reduced order model with few selected degrees of freedom is used."

>>> In Section 4.2, we highlight the current computational time of the digital twin:

"The state estimation is currently 10 times faster than real time. The virtual sensing step is twice as fast as real time, but computational improvements are possible, in particular, by using a compiled language instead of Python."

>>> Currently OpenFAST cannot be used for real time estimations as it would require an interface where the inputs and states of OpenFAST can be modified at each time step. Yet, we agree with you that some idea of the computational time for an OpenFAST simulation could give some perspective for the current work. The following was added in Section 4.2:

" For reference, OpenFAST simulations of the full TetraSpar model (with substructure flexibility) typically run 3 times slower than real time, and a reduced-order OpenFAST model with 8 degrees of freedom runs 1.1 times slower. Currently, real time estimation cannot be achieved with OpenFAST. Reduced-order modeling techniques, such as the ones presented in this article, are necessary to implement an online digital twin. Yet, if the digital twin is run as a postprocessing step, then parallelization using multiple CPUs could be used, e.g., processing different time periods of the day."

There are also questions on the hydrodynamic modeling of the floating platform. There seems to be an exclusion of the radiation damping of the platform. I assume this is due to the diameters relation to encountered wave lengths, but perhaps it would be helpful to explain this. If the platform is being modeled with Morrison's element, a brief discussion of this would be enlightening as well.

>>> Thank you for mentioning this. It is true that the original version of the manuscript was lacking details on the hydrodynamic modeling. We have added precisions at multiple places in the manuscript. In particular, we now explicitly mentioned the following:

"All the members of the substructure are modelled using the strip-theory approach (Morison equation) because the inherent long-wavelength assumption of the strip-theory approach has been shown to be sufficiently accurate for this structure with relatively slender members."

As a final statement about the hydrodynamics, what would be the effect of including wave excitation into the linearized model inputs, and why was this not used to begin with?

>>> You are correct that including wave excitation into the model could improve the modeling. It was our original intention to include it. OpenFAST currently cannot linearize with respect to wave elevation for strip-theory-based models, and one of the motivation for the python implementation pHydroDyn was to obtain such linear models. When scoping for this work and deciding to focus on tower loads, it appeared that including a wave model would not be as important because the tower loads are driven more by the wind input and platform motions (which are fairly well measured). We therefore postponed this to future work (mentioning "adding a model to account for wave excitation forces" in our future section of the conclusion). We do believe that improvements are possible by including a model for wave excitation. We believe that such modeling is important for fixed-bottom digital twins, or, for floating offshore digital twins that intend to determine substructure loads. We have added the following in the text:

"In this work, $\delta f_h$ is mapped into the inherent model noise of the Kalman filter (see Section 3.5), but further improvements could be obtained by including a model for the wave-excitation loads. In our application (tower section loads), such modeling did not appear necessary, but it will be considered in future work as it can be relevant to estimate substructure loads."

Comments/Questions:

29-30 : You mention the use of data driven models for digital twins, what are the strengths and weaknesses of these data driven models when compared to physics based? If any of these weaknesses led to the development of the physics based digital twin, it would be interesting to know.

>>> This is a fair point that we didn't highlight in the article. There are advantages and weaknesses to both approaches. We are now mentioning the main ones in the introduction:

"Machine-learning approaches can be trained using high-fidelity models or measurements, leading to potentially high accuracies while maintaining low computational time, but their training requirements imply that a technology cannot be readily transferred from one platform to another. Physics-based models often require low-fidelity models to achieve computational times low enough for digital twins to run in real time. They nevertheless offer the advantage that they provide tractable and insightful results, and they can be applied to a same family of wind turbine concepts because they do not require a training dataset. Currently, there is no definite case as to which approach can lead to the best digital twin implementation, and it is possible that future approaches will combine physics-based with data-driven techniques."

65-66: This is an important statement, as there are multiple definitions for digital twins so clarification is needed to the definition of digital twin in the scope of this article.

>>> Thank you for your comment, we hope that we made clear the definition we have in mind for this article in section 2.1

72: The goal is to use currently available measurements on the turbine (SCADA), but are there significant effects to adding more measurements? If the effect of adding more measurements to the physical turbine is positive enough, it may warrant a new addition of sensors to turbines in the field. The cost to add more sensors might be allowable if it can reduce the long term cost of the project with O&M decisions coming from digital twin estimations.

>>> Thank you for your comment, with which we strongly agree. The question of optimal placement and selection is outside the scope of this work, but we agree that it should be mentioned. We have taken some of your wording and added the following sentence in the paragraph:

"In this work, we leave open the question as to whether the installation of an additional set of optimally placed and selected sensors can further improve the predictions of the digital twin, further reducing the long-term O&M costs, and thereby warranting the additional costs of adding the sensors."

102: What values were tuned based on measurements?

>>> The OpenFAST model was implemented based on data provided by the partners. The controller was turned based on the SCADA measurements, and we have added the following sentences to provide more clarity on the controller tuning:

"The OpenFAST model is complemented with NREL's Reference OpenSource Controller (ROSCO, Abbas et al. (2021)). The controller parameters are tuned so that OpenFAST simulations match the operating conditions of the turbine extracted from SCADA data (pitch, rotor speed and power). The nacelle velocity feedback option of ROSCO is used to reduce the platform pitching motion. Using trial and error, the frequency and damping ratio of the pitch PI-controller are set to $\omega_p$ = 0.05 rad/s and $\zeta_p$ = 7\%, and the values for the torque controller are set $\omega_Q$ = 0.15 rad/s and $\zeta_Q$ = 7\%. The gain-scheduling of the pitch controller are obtained using the tuning feature of ROSCO. We note that the controller is only needed to perform verifications of the digital twin with realistic time series of the turbine responses, but the controller itself is not used for the design of the digital twin."

The method for calculating excitation forces in the OpenFAST simulation is not apparent also.

>>> We have added the following sentence to clarify the modeling of the wave excitation force:

"All the members of the substructure are modelled using the strip-theory approach (Morison equation) because the inherent long-wavelength assumption of the strip-theory approach has been shown to be sufficiently accurate for this structure with relatively slender members."

139: You mention WELIB has some shortcomings, but don't mention what they are. These options could lead to discrepancies when comparing models. Are these limited options for structural dynamics, hydrodynamics, control?

>>> Thank you for your comment, which was also pointed out by the first reviewer. The main differences lay in the structural dynamics and choice of transformation matrices (when degrees of freedom introduce "rotations"). We have reworked the paragraph to attempt to add clarifications. In particular, we have added a paragraph highlighting the differences:

"Currently, no controller or aerodynamic module is present in WELIB. Therefore, nonlinear timestepping simulations with WELIB are limited to free-decay simulations or prescribed loads. Another shortcoming is that WELIB does not cover the full range of options available with OpenFAST, which is a continuously evolving, extensively verified and validated tool. Such options include the potential flow representation of hydrodynamic bodies, the flexibility of the floating structure, aerodynamic and control features. One benefit of WELIB over OpenFAST is the possibility to perform interactive time-stepping, that is, to change the states and inputs dynamically during the simulation. We do not use this approach in this work, but it can be considered for nonlinear digital twin applications, for instance, using an extended Kalman filter algorithm. Another benefit is the possibility to obtain analytical linear models of the structure, which avoids using finite-differences and therefore reduces the associated numerical errors. In the WELIB approach, the individual modules are linearized separately before being combined into the final linear model, and it is therefore easier to understand where each term in the Jacobians of the linear models comes from, and thereby, gain physical intuitiveness on the model. Ultimately, the linear models obtained by both approaches are similar and differ mostly based on differences in the structural dynamics equations and the implementation of rotational transformation matrices."

169-175: It is not explicitly stated which method was used.

>>> Thank you for pointing that out. We have now rewritten the paragraph describing the hydrodynamic linearization and we are now precise that the one used is the python rigid-body linearization.

"The consistency between the different approaches was verified, and because of its ease of use, the second approach is retained in this study. We note that in this study, all members are modeled using the Morison equation and the hydrodynamic drag is set to zero during the linearization process."

192-194: This seems like a powerful use of the state-estimator, that is, to neglect forces and allow the estimator to account for the absence of these forces. To what degree is this valid?

>>> Thank you for this valid question. We have now tried to discuss the "degree of validity" and rationale in the text with the following addition:

"Assuming that the loading is part of the model noise is a crude approximation that is expected to be fair as long as the loading has a zero mean value, which is expected to be the case for the wave loading, but not for the wind or current loading (here omitted). This modeling choice is not very influential in this work because the motions of the platform measured by the inclinometers and GPS sensors inherently carry information about the wave loading."

322: You state the linear model is valid while close to the operating point. What is this region for a full scale turbine?

>>> This is a difficult question, and we haven't implemented gain scheduling to see if performance is improved with more linear models. We have added the following in the text:

"In this work, we used one operating point only and obtained results with fair accuracy (see Section 4). We nevertheless expect that to better represent the different operating regions of a pitch-regulated wind turbine, three to five linear models, stitched together through gain scheduling, would be needed."

356: The WELIB library is chosen for calculation of the sectional loads, but the full digital twin uses both WELIB and OpenFAST linearization. Is the job of OpenFAST linearized model to calculate the states of the system, while the WELIB calculates the section loads?  If this is not the case, clarification is needed on the responsibility of each model.

>>> You are correct, after iterating on using either linear models from WELIB or OpenFAST we have now settled to use OpenFAST for the linear model. The comparison with WELIB is yet highly instructive as it can somehow give an "origin" for each of the numerical values found with OpenFAST because we have access to the analytical linear model, and the individual mooring and hydrodynamics matrices.

For the validation results, WELIB is only used for the "virtual sensing" step. We admit that this is confusing. The following paragraph was added early on (as soon as OpenFAST and WELIB are presented) to make it clearer to the reader.

[revised manuscript text omitted]

$$M_{D0} + \delta M_D = M_{S0} + \delta M_S + s_0 \times F_{S0} + s_0 \times \delta F_S + \delta u_S \times F_{S0} + \delta u_S \times \delta F_S \tag{B13}$$

Making use of Equation B8, neglecting the nonlinear term ($\delta u_S \times \delta F_S$) and reintroducing the tilde notation, leads to:

$$\delta M_D = \delta M_S + \tilde{s}_0 \delta F_S - \tilde{F}_{S0} \delta u_S \tag{B14}$$

The Jacobians of the moments at the undisplaced node $D$ with respect to the displacements at node $D$ are then obtained by applying the
chain rule to Equation B14:

$$
\frac{\partial M_D}{\partial u_D} = \frac{\partial M_S}{\partial u_S}\frac{\partial u_S}{\partial u_D} + \frac{\partial M_S}{\partial \theta_S}\frac{\partial \theta_S}{\partial u_D} + \tilde{s}_0\left[\frac{\partial F_S}{\partial u_S}\frac{\partial u_S}{\partial u_D} + \frac{\partial F_S}{\partial \theta_S}\frac{\partial \theta_S}{\partial u_D}\right] - \tilde{F}_{S0}\frac{\partial u_S}{\partial u_D}
$$

$$
= \frac{\partial M_S}{\partial u_S} + \tilde{s}_0\frac{\partial F_S}{\partial u_S} - \tilde{F}_{S0} \tag{B15}
$$

and

$$
\frac{\partial M_D}{\partial \theta_D} = \frac{\partial M_S}{\partial \theta_S}\frac{\partial \theta_S}{\partial \theta_D} + \frac{\partial M_S}{\partial u_S}\frac{\partial u_S}{\partial \theta_D} + \tilde{s}_0\left[\frac{\partial F_S}{\partial \theta_S}\frac{\partial \theta_S}{\partial \theta_D} + \frac{\partial F_S}{\partial u_S}\frac{\partial u_S}{\partial \theta_D}\right] - \tilde{F}_{S0}\frac{\partial u_S}{\partial \theta_D}
$$

$$
= \frac{\partial M_S}{\partial \theta_S} - \frac{\partial M_S}{\partial u_S}\tilde{s}_0 + \tilde{s}_0\frac{\partial F_S}{\partial \theta_S} - \tilde{s}_0\frac{\partial F_S}{\partial u_S}\tilde{s}_0 + \tilde{F}_{S0}\tilde{s}_0 \tag{B16}
$$

**Jacobian relationships at the undisplaced destination point**

Equation B11, Equation B16, and Equation B15 can be gathered in matricial form to relate the different Jacobians between the source point
and the undisplaced destination point:

$$
\begin{bmatrix} \frac{\partial F_D}{\partial u_D} & \frac{\partial F_D}{\partial \theta_D} \\ \frac{\partial M_D}{\partial u_D} & \frac{\partial M_D}{\partial \theta_D} \end{bmatrix}_{\text{undisplaced}} = \begin{bmatrix} I & 0 \\ \tilde{s}_0 & I \end{bmatrix} \begin{bmatrix} \frac{\partial F_S}{\partial u_S} & \frac{\partial F_S}{\partial \theta_S} \\ \frac{\partial M_S}{\partial u_S} & \frac{\partial M_S}{\partial \theta_S} \end{bmatrix} \begin{bmatrix} I & -\tilde{s}_0 \\ 0 & I \end{bmatrix} + \begin{bmatrix} 0 & 0 \\ -\tilde{F}_{S0} & \tilde{F}_{S0}\tilde{s}_0 \end{bmatrix} \tag{B17}
$$

**Moments at the displaced destination point**

In this section, the moments are transferred to the displaced destination point. The vector from the displaced destination point to the displaced
source is:

$$r = s_0 + \delta u_S - \delta u_D = s_0 - \tilde{s}_0\delta\theta_S \tag{B18}$$

Introducing Equation B6 and Equation B18 into Equation B2, and temporarily using the "×" notation instead of the tilde notation:

$$M_{D0} + \delta M_D = M_{S0} + \delta M_S + s_0 \times F_{S0} + s_0 \times \delta F_S - (s_0 \times \delta\theta_S) \times F_{S0} - (s_0 \times \delta\theta_S) \times \delta 
[revised manuscript text omitted]

874 469-2017, 2017.

---

## Referee Report (RR1)

Review of the paper "A digital twin solution for floating offshore wind turbines validated using a full-scale prototype".

The paper deals with the implementation of a physics-based digital twin of an offshore turbine and its validation using data coming from a full-scale prototype.

The Author did a great job building up the physics-based twin emphasizing the fact that this task is extremely difficult and requires a proper implementation of different estimators (wind speed estimator, thrust estimator, loads estimator, …). In this work, which is still preliminary, the Authors relied mostly on Kalman filtering. Given the sophistication of the twin and the inherent complexity of the system, the agreement in the tower damage equivalent load estimation of lower than 15% is an excellent result.

I had also the opportunity to check the reviews of the previous round and the replies. It seems that the Authors provided fully adequate responses and corrections to the manuscript.

For this reason, I recommend publishing the paper.

I have only three minor requests that I hope the Authors may consider.

1. Abstract, line 12: it would be fair to add that the estimated errors of 10% refer to the tower fore-aft damage equivalent loads.
2. Model, section 2.3.3: why did the Authors consider only the fore-aft tower bending mode, when the side-side is also important as it is low-damped?
3. Linearization of the system: I would suggest that the linearization of the entire system could be done in multiblade coordinates to also capture the effect of the periodicity of the system. For example, whirling modes, that affect tower loads and fatigue, are only captured after applying MB transformation.

---

## Author Response (AR2)

Dear reviewers,

Thank you so much for your time and effort in reviewing our revised version of the paper. Please find our answers to your comments and the corresponding updates to the manuscript below. We hope to have addressed most of your comments which helped us improve the manuscript, and we will be happy to continue the discussion. In the coming days, we will submit the revised manuscript.

Emmanuel and co-authors

**Reviewer 1**

1. Abstract, line 12: it would be fair to add that the estimated errors of 10% refer to the tower fore-aft damage equivalent loads.
> Thank you for noting this. We agree, and have added this precision in the text.
In this simulation realm, we obtain estimated damage equivalent loads of the tower fore-aft bending moment with an accuracy of approximately 5% to 10%.

2. Model, section 2.3.3: why did the Authors consider only the fore-aft tower bending mode, when the side-side is also important as it is low-damped?
> This is a fair point and we agree that this degree of freedom should be added in the future. We didn't include the side-side for simplicity (even though he effort to add it is not too large). One extra complication is that YAMS and OpenFAST handle rotations differently, and having two rotations at the tower top would lead to differences in results between the two frameworks. We've added the following to the text (not diving into details, but your point about the low damping makes a lot of sense):
"The side-side tower bending can be added in a similar way, but for simplicity, it was not considered in this study."

3. Linearization of the system: I would suggest that the linearization of the entire system could be done in multiblade coordinates to also capture the effect of the periodicity of the system. For example, whirling modes, that affect tower loads and fatigue, are only captured after applying MB transformation.
> This is a good point that would probably require some thoughts as to how to apply our methodology for more general cases. In our case, we do not have degrees of freedom for the blades or the shaft bending, therefore we didn't have to worry about the quantities being expressed in the rotating frame of reference. We can still expect some periodicity of the state matrix due to tilting. We had look at this in the past and found little variation of the A matrix with azimuth, and a simple averaging over different azimuthal values was seen to give good steady state linear model. To your point, MB would indeed be required if the model where to use states, inputs and outputs that are specified for each blade. We did not add anything in the text as we would probably need to investigate this issue more to see how it would affect our methodology.